Methods
# Concatemer-assisted stoichiometry analysis: targeted mass spectrometry for protein quantification

Jiaxi Cai[1,2] , Yun Quan[1] , Cindy Yuxuan Zhang[1], Ziyi Wang[1] , Stephen M Hinshaw[3] , Huilin Zhou[1,2,4] , Raymond T Suhandynata[5,6]

**Large multiprotein machines are central to many biological processes. However, stoichiometric determination of protein complex subunits in their native states presents a significant challenge. This study addresses the limitations of current tools in accuracy and precision by introducing concatemer-assisted stoichiometry analysis (CASA). CASA leverages stable isotope-labeled concatemers and liquid chromatography–parallel reaction monitoring–mass spectrometry (LC-PRM-MS) to achieve robust quantification of proteins with sub-femtomole sensitivity. As a proof of concept, CASA was applied to study budding yeast kinetochores. Stoichiometries were determined for ex vivo reconstituted kinetochore components, including the canonical H3 nucleosomes, centromeric (Cse4[CENP-A]) nucleosomes, centromere proximal factors (Cbf1 and CBF3 complex), inner kinetochore proteins (Mif2[CENP-C], Ctf19[CCAN] complex), and outer kinetochore proteins (KMN network). Absolute quantification by CASA revealed Cse4[CENP-A] as a cell cycle–controlled limiting factor for kinetochore assembly. These findings demonstrate that CASA is applicable for stoichiometry analysis of multiprotein assemblies.**

## Introduction

Stoichiometric analyses and absolute quantifications of native protein complexes have been challenging considering the complexity of native biological samples and the precision required to confidently determine protein ratios within twofold (e.g. 3:2 or 4:3). To address this, we present concatemer-assisted stoichiometry analysis (CASA) and its application to study one of the most intricate multiprotein assemblies of the cell—the kinetochore.

In eukaryotes, kinetochores ensure the faithful segregation of chromosomes during cell division and act as the load-bearing junctions between centromeric chromatin and spindle microtubules. Despite considerable protein sequence divergence across the eukaryotic kingdoms, kinetochores have a broadly conserved structural organization and dozens of functionally conserved subunits (Biggins, 2013; Musacchio & Desai, 2017). Kinetochores comprise two subregions: the inner and outer kinetochores, each containing multiple protein subcomplexes. Inner kinetochore proteins assemble on chromosomal centromeres, whereas outer kinetochore proteins build upon the inner kinetochore and bind microtubules. Because of this structural and functional conservation, budding yeast *Saccharomyces cerevisiae* has been extensively used as a model organism for characterizing kinetochore structures and functions. Prior studies have identified most, if not all, kinetochore subunits in yeast (Meluh & Koshland, 1995; Ortiz et al, 1999; Cheeseman et al, 2002; Measday et al, 2002; Westermann et al, 2003; Cohen et al, 2008; Lawrimore et al, 2011; Bock et al, 2012; Schleiffer et al, 2012; Hornung et al, 2014). Structures of many kinetochore subcomplexes have also been determined (Cho & Harrison, 2011; Hinshaw & Harrison, 2013, 2019, 2020; Yan et al, 2018, 2019; Leber et al, 2018; Hinshaw et al, 2019; Guan et al, 2021; Dendooven et al, 2023; Deng et al, 2023). Moreover, dynamic interactions between centromeres, kinetochore proteins, and microtubules have been examined through fluorescent microscopy (Joglekar et al, 2006; Kitamura et al, 2007; Campbell & Desai, 2013; Wisniewski et al, 2014; Dhatchinamoorthy et al, 2017; Li et al, 2023).

Despite these advances, the budding yeast kinetochore has yet to be fully reconstituted using recombinant proteins. For instance, Mif2, the budding yeast ortholog of human CENP-C, is essential for kinetochore assembly (Meluh & Koshland, 1995; Cohen et al, 2008; Klare et al, 2015) but has yet to be included when reconstituting kinetochores in vitro (Dendooven et al, 2023). Post-translational modifications of Mif2[CENP-C] control inner kinetochore assembly (Hagemann et al, 2022 *Preprint*; Hinshaw et al, 2023; Klemm et al, 2023 *Preprint*); however, our understanding of the regulatory mechanism is still incomplete. Cse4, the yeast ortholog of human CENP-A, is a histone H3 variant specific to centromeric chromatin (Biggins, 2013; Deng et al, 2023). Despite reports of biochemical Cse4-centromere reconstitutions (Xiao et al, 2017; Yan et al, 2019),

[1]Department of Cellular and Molecular Medicine, University of California, San Diego, San Diego, CA, USA    [2]Department of Bioengineering, University of California, San Diego, San Diego, CA, USA    [3]Department of Chemical and Systems Biology, Stanford University, Palo Alto, CA, USA    [4]Moores Cancer Center, University of California, San Diego, San Diego, CA, USA    [5]Skaggs School of Pharmacy and Pharmaceutical Sciences, University of California, San Diego, San Diego, CA, USA    [6]Department of Pathology, University of California, San Diego, San Diego, CA, USA

Correspondence: huzhou@health.ucsd.edu; rtsuhandynata@health.ucsd.edu

capturing this complex in quantities suitable for structural or mechanistic studies has been a major challenge. Recently reported efforts include the use of a single-chain antibody fragment (Guan et al, 2021) and the use of a chimeric Widom-601-centromere III fusion DNA fragment (Dendooven et al, 2023) to stabilize the association between the Cse4[CENP-A]-nucleosome, CEN DNA, and the essential CBF3 complex. Perhaps this is not surprising, considering that the chaperone Scm3[HJURP] is required for assembling the Cse4[CENP-A]-nucleosome in cells, along with the CBF3 complex that directly recognizes the native centromere (Hyman et al, 1992; Sorger et al, 1994; Camahort et al, 2007; Mizuguchi et al, 2007; Stoler et al, 2007; Yan et al, 2018). Therefore, the mechanism and end-state product describing the assembly of the Cse4[CENP-A] nucleosome on native centromeres have yet to be fully understood.

Considerable progress has been made toward reconstituting kinetochores on native centromeric DNA by leveraging the well-defined DNA sequences of budding yeast's point centromeres in conjunction with concentrated yeast whole-cell extracts (Kingsbury & Koshland, 1991; Sorger et al, 1994; Sandall et al, 2006; Lang et al, 2018). The ex vivo kinetochore reconstitution system mirrors the physiological requirements for native kinetochore assembly, including the conserved CDE III sequence and the presence of the CBF3 complex (Lang et al, 2018). Single-molecule studies have begun to reveal biophysical properties of ex vivo reconstituted kinetochores (Popchock et al, 2023). However, kinetochore subunit stoichiometry in this system has yet to be determined because of the low abundance of kinetochores assembled via this approach. The kinetochore problem highlights the need for sensitive and robust protein quantification in biological samples, inspiring the development of CASA, a targeted LC-MS/MS protein quantification platform.

LC-MS/MS has been widely used to analyze peptides and proteins. In most cases, untargeted LC-MS/MS approaches are used to identify and quantify peptides/proteins, particularly when peptides/proteins of interest are unknown (Domon & Aebersold, 2010). When the analytes of interest are known, targeted LC-MS/MS approaches such as multiple reaction monitoring (MRM) and parallel reaction monitoring (PRM) have been shown to provide superior selectivity and sensitivity (Sherwood et al, 2009; Gallien et al, 2012; Peterson et al, 2012; Bourmaud et al, 2016; Lawrence et al, 2016; Hoffman et al, 2018). Improvements in quantitative accuracy and precision are especially apparent when targeted LC-MS/MS approaches are combined with isotope dilution using a stable isotope-labeled internal standard (Gerber et al, 2003; Geiger et al, 2011). There are several approaches for generating the stable isotope-labeled internal standard to facilitate absolute quantification of proteins, such as synthesis of isotope-labeled peptides, metabolic protein labeling, and quantitative concatemers (i.e., QconCAT) (Pratt et al, 2006). Specifically, quantitative concatemers have been successfully adopted for biological applications to quantify multiple proteins in a single experiment (Takemori et al, 2017), making them ideal for monitoring larger protein complexes.

To develop a method useful for quantifying subunit abundance in a single large protein complex, a kinetochore-specific concatemer was produced by stable isotope labeling by amino acids in cell culture (SILAC) (Ong et al, 2002) and purified from a budding yeast expression system. For simplicity, this concatemer is referred to as the concatenated kinetochore protein (CKP). The CKP is comprised of tryptic

peptides derived from over 20 kinetochore subunits and was used as the stable isotope-labeled internal standard for a targeted liquid chromatography–parallel reaction monitoring–mass spectrometry (LC-PRM-MS) method. The analytical performance of CASA for kinetochore quantification was evaluated, revealing its linearity, sensitivity, accuracy, and precision. CASA was then used to study ex vivo reconstituted kinetochores as a proof of concept. The development and application of CASA not only serve the purpose of validating and expanding our existing knowledge of the yeast kinetochore but also hold the potential for facilitating accurate quantitative analysis of other protein complexes.

# Results

## Design of the CKP

The CKP contains 25 concatenated tryptic peptides derived from kinetochore subunits (Fig 1A and B), whose sequences are listed in Table 1. CKP expression was performed in yeast using a high-copy expression plasmid with a strong galactose-inducible promoter to maximize yield. C-terminal arginine peptides were selected to facilitate the incorporation of isotopically light ($^{12}C_6$ $^{14}N_4$) or heavy arginine ($^{13}C_6$ $^{15}N_4$) via the SILAC approach in an arginine auxotrophic yeast strain (Chen et al, 2010; Albuquerque et al, 2013). To promote robust protein expression, stability, and ease of purification, the CKP is flanked by N-terminal GST and C-terminal 6xHIS-3xFLAG tags. The N-terminal GST, a globular protein, aids with CKP solubility during its expression, whereas the C-terminal tags were used to purify the CKP. Expression and purification of both isotopically light CKP (L-CKP) and isotopically heavy CKP (H-CKP) were evaluated by anti-FLAG Western blotting or Coomassie staining (Fig S1). The Western blot and Coomassie staining confirmed that full-length CKPs were purified, ensuring that tryptic peptides of the kinetochore are iso-stoichiometric within the CKPs.

Next, PRM scans of both light and heavy tryptic peptides of the CKP were incorporated into an initial targeted LC-MS/MS method using a normalized collision energy (NCE) of 27%. Using this PRM method, extracted product ion chromatographs of H-CKP peptides (Fig 1C) confirmed the thorough incorporation of the heavy stable isotopes (>99%) with undetectable signal observed for PRM scans of the corresponding light peptides (Table S1). Detection sensitivity by PRM-MS differs according to peptide sequence and corresponding chemical properties, leading to differences in observed product ion intensities (Fig 1C).

Source data are available for this table.

## Evaluation of peptide collision energy

Collision energy (CE) was evaluated for all 25 peptides of the CKP (Fig 2) on-column via replicate injections. Collision energy (CE) or NCE was increased in series by 2% for each injection until product ion signal intensities decreased. The resulting breakdown curves for two representative peptides are shown in Fig 2A and B. Gaussian distributions were observed for the Mcm21 peptide (IDDISTSDR) and the Nkp1 peptide (EIYDNESELR) between 10% and 22% CE (Fig 2A and

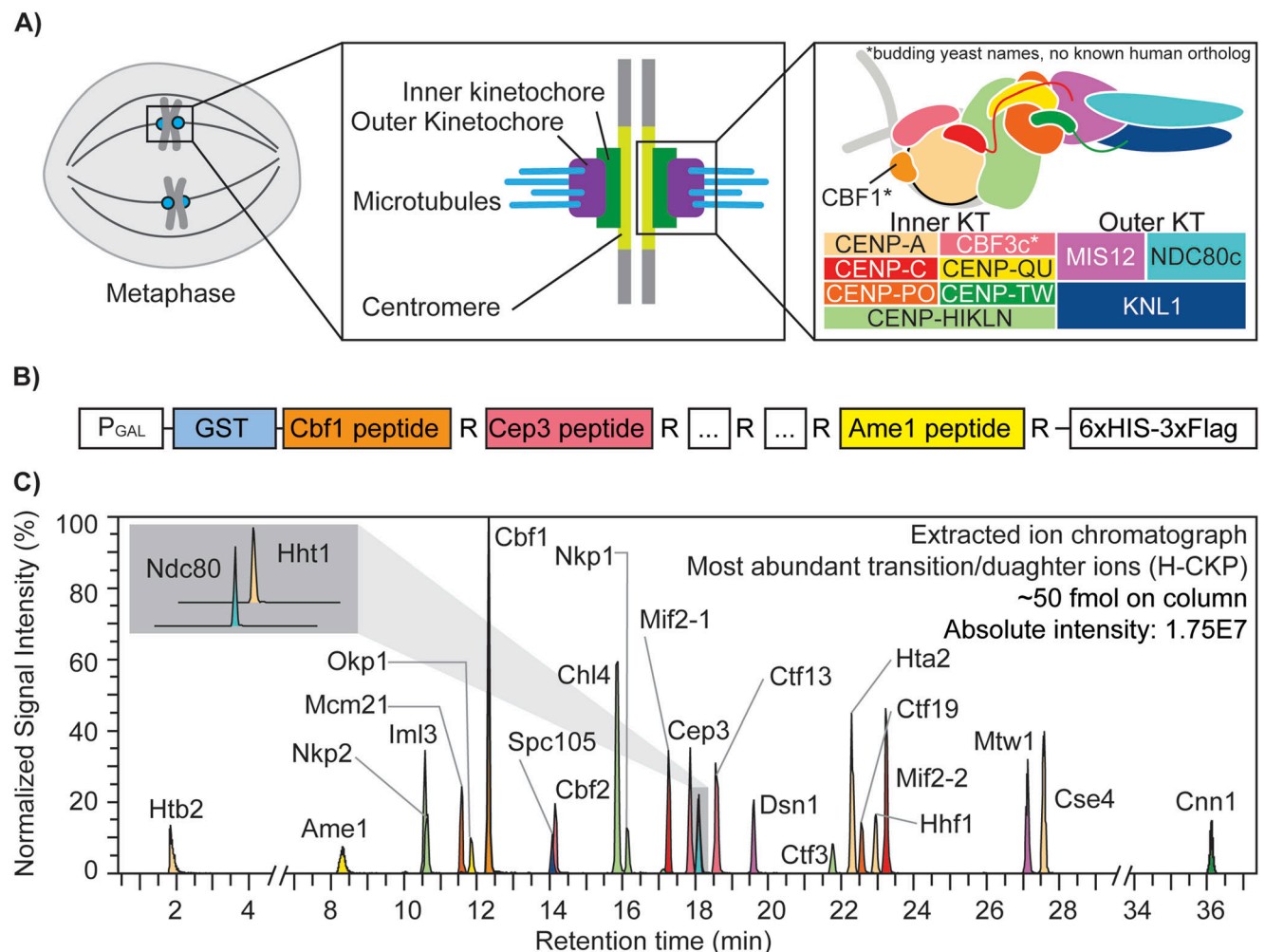

**Figure 1. Kinetochore and the concatenated kinetochore protein (CKP) construct.**
**(A)** Schematics of yeast kinetochore organization: yeast kinetochores assemble on the point centromere at each chromosome to carry out mitotic chromosome segregation. The kinetochore can be divided into inner and outer subcomplexes, where the inner kinetochore contacts the centromeric chromatin, and the outer kinetochore forms an association with the spindle microtubules. Both inner and outer kinetochores are composed of multiple protein complexes, as illustrated and annotated in the right-most panel. **(B)** Gene block is constructed to express and purify the CKP: 25 tryptic peptides with a C-terminal arginine were selected from DDA-MS results and incorporated into the CKP. The expression of the protein is controlled by a galactose-inducible promoter ($P_{GAL}$) in yeast. **(C)** Extracted ion chromatograph of the most abundant daughter ions of H-CKP peptides with retention time in minutes. The y-axis indicates the signal intensities of each ion expressed as a percentage of the signal from the ion with the highest signal intensity (Cbf1 peptide). The total amount of H-CKP injected and the absolute signal intensity for the Cbf1 peptide's most abundant product ion are labeled on the top right of the plot.

B). Optimal CEs of all peptides were determined based on the overall signal intensities of the top product ions. CE fragmentation produced greater changes in product ion signal intensity relative to NCE fragmentation, and the highest product ion signal intensities were observed when using the CE fragmentation scheme for all peptides but the Htb2 peptide (Table 2).

Signal intensities of each peptide's top 10 most abundant ions were compared before and after CE optimization (Fig 2C). Optimal CEs were compared with NCE at 27%, a commonly used CE for proteomics analysis on similar instrumentation (Lawrence et al, 2016). Improved signal intensities were observed for 21 peptides, and no improvements were observed for the Dsn1, Spc105, Cnn1, and Cse4 peptides (Fig 2C). The relative ion intensity among product ions (ion ratio) is unique to a peptide of a defined amino acid sequence at a given collision energy.

Thus, having diverse product ions allows for the selection of different product ions as quantifiers for quantification and qualifiers to confirm identification and resolve interferences. PRM-MS monitors all daughter ions during the evaluation of CE and NCE, which allowed us to revise the optimal NCE/CEs after taking into consideration matrix interferences and quantifier ion signal intensities for the final PRM-MS method (Table S2).

### Peptide stability and matrix stabilization of peptides

Because of the ~1-h duration of the LC-PRM-MS method, an analytical batch can take over 24 h to complete, and peptide digests are kept at 10°C in the autosampler for this duration. Thus, peptide stability was evaluated by analyzing replicate injections (n = 6) of

**Table 1.   Peptides of the CKP.**

| Protein | Peptide | Mass-to-charge ratio (m/z) (light) | Charge | Retention time window | |
|---|---|---|---|---|---|
| | | | (z) | Start (min) | End (min) |
| Cbf1 | LSTEDEEIHSAR | 462.8880 | 3 | 10.73 | 14.73 |
| Cep3 | LVYLTER | 447.2582 | 2 | 16.23 | 20.23 |
| Ctf13 | TGLADFTR | 440.7298 | 2 | 17.04 | 21.04 |
| Cse4 | YTPSELALYEIR | 727.8799 | 2 | 25.23 | 29.23 |
| Htb2 | HAVSEGTR | 428.7172 | 2 | 1.00 | 3.94 |
| Hta2 | AGLTFPVGR | 459.2638 | 2 | 20.38 | 24.38 |
| Hhf1 | ISGLIYEEVR | 589.8244 | 2 | 20.96 | 24.96 |
| Hht1 | STELLIR | 416.2504 | 2 | 16.51 | 20.51 |
| Mif2-1 | YSLDTSESPSVR | 670.8201 | 2 | 15.61 | 19.61 |
| Mif2-2 | VAPLQYWR | 516.7849 | 2 | 21.20 | 25.20 |
| Cbf2 | EENIVNEDGPNTSR | 787.3581 | 2 | 12.62 | 16.62 |
| Okp1 | VIQAEYR | 439.7402 | 2 | 10.45 | 14.45 |
| Ame1 | NDEDLTTR | 482.2225 | 2 | 7.35 | 11.35 |
| Mcm21 | IDDISTSDR | 511.2435 | 2 | 10.24 | 14.24 |
| Ctf19 | QQLSLLDDDQVR | 715.3677 | 2 | 20.67 | 24.67 |
| Ctf3 | DAPGSATLILQR | 621.3461 | 2 | 19.90 | 23.90 |
| Iml3 | ESIVTSTR | 446.7404 | 2 | 9.37 | 13.37 |
| Chl4 | NEDSGEPVYISR | 683.3177 | 2 | 14.25 | 18.25 |
| Cnn1 | SFLQDLSQVLAR | 688.8803 | 2 | 33.32 | 37.32 |
| Nkp1 | EIYDNESELR | 634.2937 | 2 | 14.63 | 18.63 |
| Nkp2 | VTSELEAR | 452.7404 | 2 | 9.37 | 13.37 |
| Ndc80 | QYDSSIQNLTR | 662.8282 | 2 | 16.37 | 20.37 |
| Dsn1 | ILDNTENYDDTELR | 855.8945 | 2 | 17.99 | 21.99 |
| Spc105 | VHISTQQDYSPSR | 506.5829 | 3 | 12.31 | 16.31 |
| Mtw1 | IPEEYLDANVFR | 733.3697 | 2 | 25.02 | 29.02 |

The peptides are listed from top to bottom in the same order as arrayed in the CKP (N- to C-terminus). The first column of the table contains the names of proteins represented by each peptide and is color-coded to match Fig 1A.

digested H-CKP kept at 10°C in the autosampler across a period of ~30 h, and recovery (peak area) for all 25 peptides reconstituted in 0.1% TFA (no matrix) was monitored (Fig 3). After 15 h, 18 peptides were recovered at >70% of initial levels, 2 peptides between 70% and 50%, and 5 peptides at <50% (Fig 3B and Table 3). At 30 h, 6 peptides were recovered at >70% of initial levels, 10 peptides between 70% and 50%, and 9 peptides at <50% (Fig 3B and Table 3). Thus, most peptides had poor recoveries across this period, indicating poor stability at 10°C. Some peptides had particularly poor recoveries, which include those derived from Cnn1, Cse4, and Mif2. Common features of these peptides include that they elute at later retention times on the LC gradient (Fig 1C) and contain a higher proportion of large hydrophobic amino acid residues such as tryptophan, phenylalanine, leucine, and isoleucine relative to other targeted peptides (see Table 1 for peptide sequences).

To address the poor peptide recovery, 50 ng/µl of digested yeast cytoplasmic proteins was included as a stabilizing matrix to circumvent losses from peptide precipitation and non-specific adsorption losses in-vial. The effect of matrix stabilization was evaluated, as shown in Fig 3, after the reconstitution of digested H-CKP in the stabilization matrix instead of 0.1% TFA. After 30 h in a 50-ng/µl stabilization matrix, all peptides were recovered at >85% of initial levels except for the Cnn1 peptide. Recovery for the Cnn1 peptide was 89.5% at 15 h and 46.1% at 30 h (Table 3). Direct comparisons of the matrix effects on peptide recoveries are illustrated in Fig 3A for three representative peptides of varying recoveries across 30 h. Thus, the stabilization matrix significantly improved the recoveries of all 25 peptides over a 30-h period (Fig 3B). The matrix was necessary for maintaining peptide stability and facilitating acceptable recoveries during subsequent method validation and data analysis.

## Evaluation of the analytical measurement range and limits of quantification

Evaluation and establishment of the analytical measurement range (AMR) were performed to ensure robust peptide quantification. To determine the AMR for each peptide, including the lower limit of

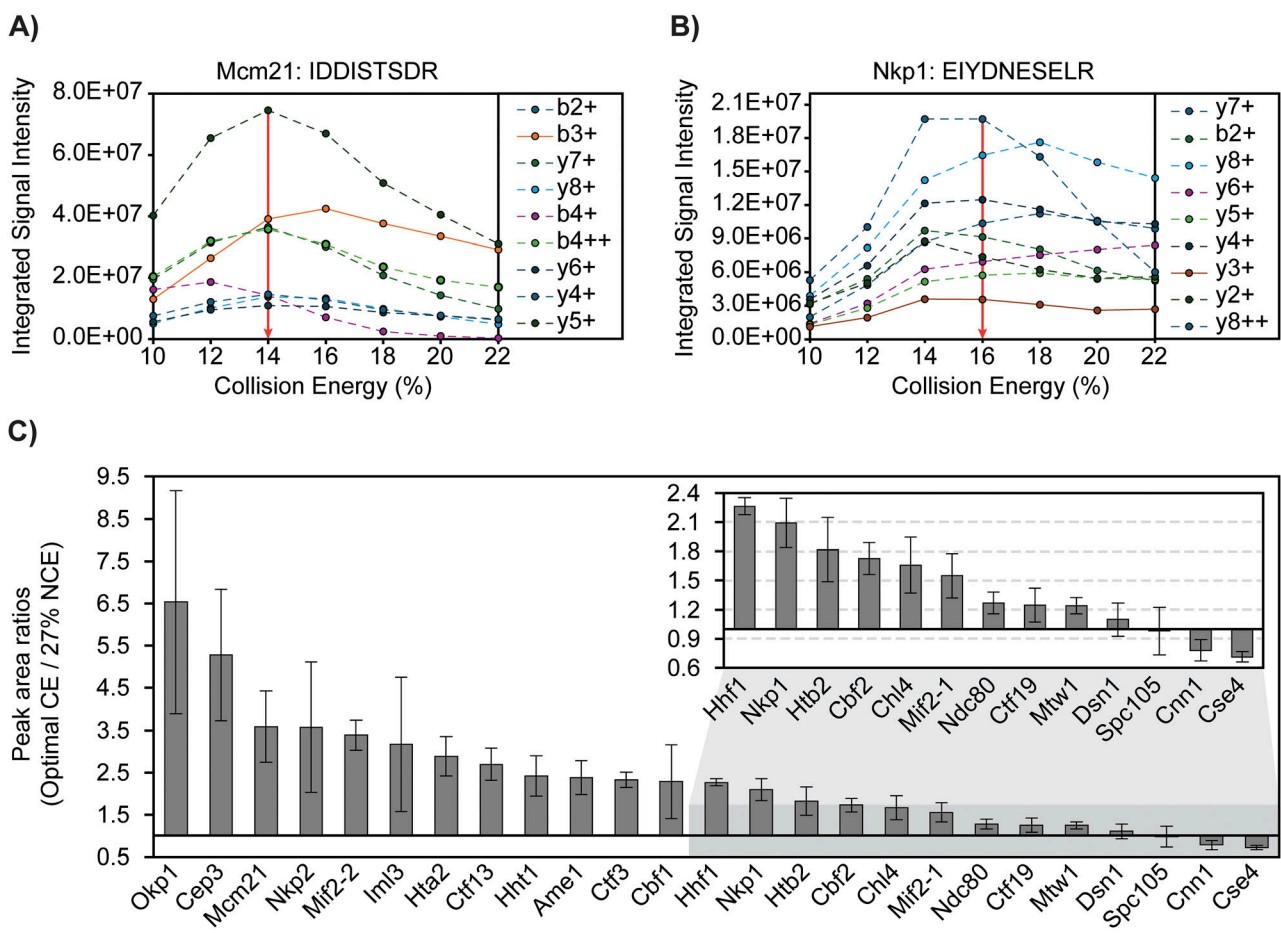

**Figure 2. Collision energy is optimized for each CKP peptide to improve signal intensity and transition heterogeneity.**
(A, B) Representative breakdown curves of IDDISTSDR (Mcm21 peptide) and EIYDNESELR (Nkp1 peptide): the peak areas of the top nine most abundant product ions were plotted against %CE to illustrate the changes in fragmentation patterns of the peptide in response to changes in collision energies. **(C)** Peak area ratios of product ions before and after CE optimization: the mean peak area of each peptide's most abundant product ions (sum of peak areas of the top 10 most abundant product ions) across three analytical replicates was plotted for injections using either 27% NCE for all peptides or optimal CEs (see Table 2). Mean peak area ratios were plotted in descending order as gray bars, and standard deviations were plotted as error bars. The x-axis intersects the y-axis at y = 1, which indicates no improvement of product ion signal intensity from CE optimization.
Source data are available for this figure.

quantification, 10 calibrators comprised of digested L-CKP with concentrations ranging from 78 pM to 60 nM in-vial were mixed with a fixed concentration of the internal standard (1 nM of H-CKP) in-vial and reconstituted in the stabilization matrix. Using the peak area ratio (analyte peak area/internal standard peak area), external calibration curves for all 25 peptides were generated with the requirement of a minimum of four calibrator levels, calibrator biases within ±20% from nominal target values (Table S3), and $R^2$ values greater than 0.99 (Table 4).

All calibrators were evaluated for inter-day accuracy and precision through analytical replicates (Table S4 for inter-day accuracy and Table S5 for inter-day precision). Product ions were chosen for peptide quantification (quantifier ions) and identification (qualifier ions) based on the signal intensity and the absence of matrix interference (Table S6 for L-CKP ion masses and Table S7 for H-CKP ion masses).

The lower limit of quantification for each peptide was determined as the lowest calibrator level with a signal-to-noise ratio greater than 10 in the quantifier ion, average biases within ±20% for accuracy, and

%CVs < 15% for precision. The AMR for each peptide was then established as the range from the lower limit of quantification up to the highest calibrator level, which met the accuracy and precision acceptability criteria stated above. In summary, the majority of peptides (84%) have AMRs spanning across two orders of magnitude (Fig 4). Fig 4A shows an example peptide with an AMR spanning 10 calibrators, and Fig 4B shows an example peptide with a much more limited AMR. Fig 4C and Table 4 summarize the AMRs of all CKP peptides. The Mtw1 and Cse4 peptides have 2 calibration curves with overlapping AMRs that comprise the reportable range shown in Fig 4C. As an example, Fig S2 shows the high- and low-range calibration curves of Cse4.

## CASA of ex vivo reconstituted kinetochores using M-phase yeast cell extracts

To demonstrate the suitability of this LC-PRM-MS method for determining protein complex stoichiometry, ex vivo reconstituted

**Table 2. Summary of optimal collision energies.**

| Protein | Peptide | (N)CE |
| --- | --- | --- |
| Cbf1 | LSTEDEEIHSAR | 26 |
| Cep3 | LVYLTER | 10 |
| Ctf13 | TGLADFTR | 12 |
| Cse4 | YTPSELALYEIR | 19 |
| Htb2 | HAVSEGTR | (20) |
| Hta2 | AGLTFPVGR | 10 |
| Hhf1 | ISGLIYEEVR | 16 |
| Hht1 | STELLIR | 10 |
| Mif2-1 | YSLDTSESPSVR | 22 |
| Mif2-2 | VAPLQYWR | 12 |
| Cbf2 | EENIVNEDGPNTSR | 25 |
| Mcm21 | IDDISTSDR | 14 |
| Ctf19 | QQLSLLDDDQVR | 24 |
| Ctf3 | DAPGSATLILQR | 16 |
| Iml3 | ESIVTSTR | 10 |
| Chl4 | NEDSGEPVYISR | 20 |
| Mtw1 | IPEEYLDANVFR | 25 |
| Cnn1 | SFLQDLSQVLAR | 14 |
| Nkp1 | EIYDNESELR | 16 |
| Nkp2 | VTSELEAR | 10 |
| Ndc80 | QYDSSIQNLTR | 23 |
| Dsn1 | ILDNTENYDDTELR | 28 |
| Spc105 | VHISTQQDYSPSR | 10 |
| Okp1 | VIQAEYR | 10 |
| Ame1 | NDEDLTTR | 12 |

Parentheses indicate NCE was used. Otherwise, CE was used.

kinetochores were analyzed (Fig 5). Ex vivo yeast kinetochores were reconstituted on biotinylated DNA-bound streptavidin beads using yeast whole-cell extracts derived from cells arrested in the M phase by nocodazole (Fig 5A). Reconstituted kinetochores were eluted from DNA beads using DNase I, and eluates were prepared for tryptic digestion and subsequent isotope dilution LC-PRM-MS analysis. The reconstitution and DNase I elution efficiency were first evaluated by Western blotting using a Mif2-TAF-tagged strain (HZY2347) (Fig S3). After this, a *bar1Δ* strain (HZY1029) was used for all subsequent reconstitutions for PRM-MS. Kinetochore reconstitutions on three different types of DNA beads were evaluated using process replicates (n = 3): CEN DNA (WT yeast centromere III), MUT DNA (yeast centromere III with a point mutation at CDE III), and ARS DNA (autonomously replicating sequence) (Fig 5A).

Canonical histones associated readily with DNA, regardless of sequence. All four histone subunits were determined to be iso-stoichiometric in all studies, except for fluctuations in the amount of Htb2 associated with MUT DNA (Fig 5B). This Htb2 peptide suffers from significant ion suppression in the reconstitution sample matrices, leading to low internal standard

recovery (Fig S4). Cbf1 was robustly reconstituted on CEN DNA and MUT DNA but not on the ARS DNA (200-fold enrichment, Fig 5B). Indeed, Cbf1 binds the E-box consensus sequence (CACGTG) in the CDE I region of the centromere (Wieland et al, 2001). The CBF3 complex (Cbf2-Cep3-Ctf13-Skp1; Skp1 was not monitored here) bound specifically to the CEN DNA and not to the MUT DNA or the ARS DNA (Fig 5B). This confirms the specific role of the CBF3 complex in recognizing the conserved CCG element of CDE III (Lechner & Carbon, 1991; Hyman et al, 1992), which was mutated to AGC in the MUT DNA. The three monitored subunits of the CBF3 complex have an average calculated integer stoichiometry of 12 : 8 : 7 (Cbf2 : Cep3 : Ctf13), which deviates from the stoichiometry of in vitro reconstituted CBF3 complexes (2 : 2 : 1; Cbf2 : Cep3 : Ctf13) (Cho & Harrison, 2011; Leber et al, 2018). However, a stoichiometry of 2 : 2 : 1 (Cbf2 : Cep3 : Ctf13) can be assigned to the complex while staying within the 95% confidence intervals of the peptide measurement (n = 3).

The centromere-specific H3 variant, Cse4, was observed to reconstitute specifically on the CEN DNA (~20-fold higher) relative to the MUT or ARS DNA (Fig 5B). Unlike histone H3, Hht1, Cse4 loading is much less efficient (~4-fold less). The amount of Cse4 on the CEN DNA was ~4-fold lower than Cbf2. Cse4 is expected to be 1 : 1 to Cbf2 when completely loaded (Guan et al, 2021), indicating that not all loaded CBF3 complexes were associated with Cse4. Likewise, the observed Cse4: Mif2 stoichiometry was ~3 : 1, indicating that not all Cse4 nucleosomes were associated with Mif2 (Fig 5C). Notably, the two peptides of Mif2 did not quantify iso-stoichiometrically, showing a deviation of 34% (Fig 5C). Analytical variability was ruled out because both Mif2 peptides were quantified reproducibly within the AMR with %CVs < 15% from three process replicates (Table S8). In addition, the calculated average integer stoichiometry of Ame1: Okp1 was 4 : 3, whereas 1 : 1 stoichiometry is expected as Ame1 and Okp1 form a stable heterodimer (Hinshaw & Harrison, 2019). However, 1 : 1 stoichiometry (Ame1 : Okp1) can be assigned while staying within their 95% confidence intervals. Though modest, such discrepancies should be addressable with additional Mif2, Ame1, and Okp1 peptides (see Discussion section). Interestingly, the stoichiometry between Mif2 and Ame1-Okp1 was close to 1 : 1, indicating that these proteins reconstituted at a comparable level on the Cse4-bound CEN DNA (Fig 5C). The outer kinetochore subunits reconstituted on centromere DNA at further reduced levels relative to the inner kinetochore subunits (Fig 5D). Finally, the remaining non-essential inner kinetochore subunits were quantified at reduced levels relative to Mif2 and Ame1-Okp1 (Fig 5E).

Altogether, under the assumption that each molecule of DNA associates with one centromeric or canonical nucleosome, we calculated that ~40% of the CEN DNA is non-specifically associated with canonical nucleosomes, ~25% is associated with Cbf1 and the Cbf3 complex, but only ~7% is associated with Cse4. Thus, Cse4 recruitment is a critical limiting factor of this ex vivo kinetochore assembly system. Subsequent assembly is not stoichiometric, as only ~2% of CEN DNA is associated with Mif2 and Ame1-Okp1 and <1% with outer kinetochore subunits (Fig 5). Thus, additional factors limit reconstitution relative to the kinetochore in vivo, where 1 Cse4 nucleosome recruits 7–8 outer kinetochore complexes (Joglekar et al, 2006). The ability to quantitatively monitor assembly

## A)

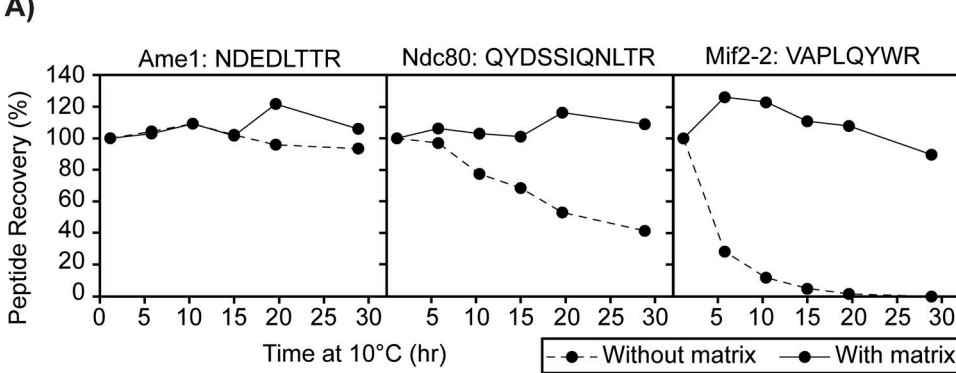

## B)

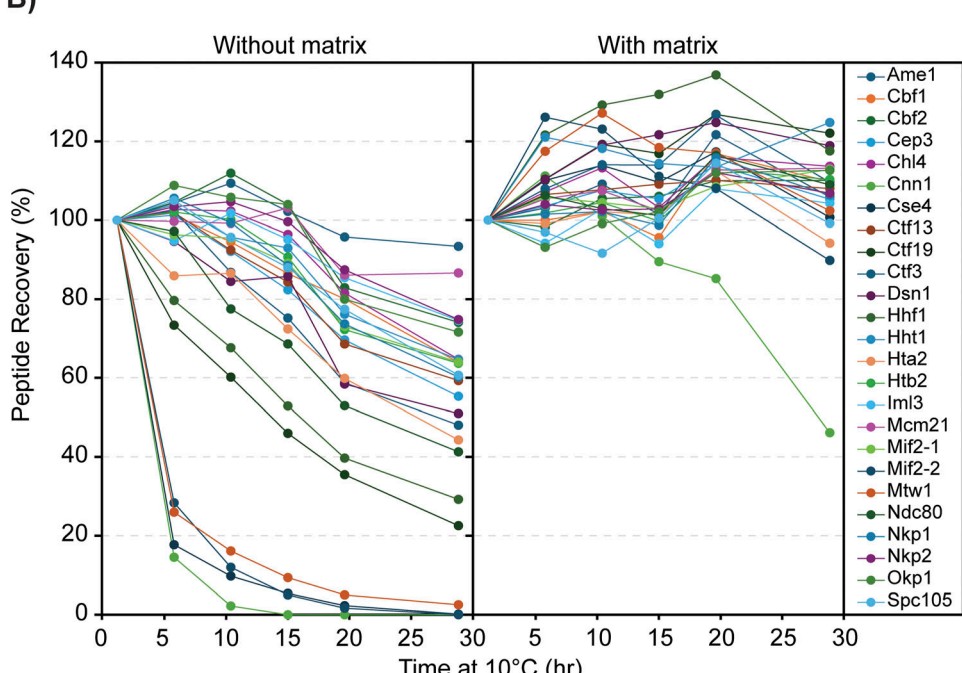

**Figure 3. Effects of stabilization matrix on peptide recovery.**
**(A)** Representative peptide recovery profiles over time: no matrix (0.1% TFA)—solid line; with matrix (0.1% TFA with 50 ng/μl digested yeast cytosolic proteins)—dashed line. The peptides' sequences are labeled on the top of each panel. The y-axis shows the percentage recovery of the peptide by peak area in comparison with the first time point (1 h at 10°C), and the x-axis shows the amount of time that the digested CKP was left at 10°C before injection into the MS. **(B)** Recovery profiles of all CKP peptides: as above, the percentage recovery of each peptide was plotted against time. Each peptide is marked by a different color, as indicated by the legend key on the right-most panel.
Source data are available for this figure.

will facilitate the development of conditions that improve the reconstitution to match the physiological assembly state.

### Effect of cell cycle on ex vivo reconstituted kinetochores

Finally, we addressed the effect of cell cycle stages on kinetochore reconstitution by comparing extracts from G1-phase cells with those from M-phase cells (Fig 6). In G1-phase cell extracts, histones associated with DNA (CEN and MUT) robustly (Fig 6A). Cbf1, CBF3, and other inner kinetochore components reconstituted specifically on CEN DNA and not MUT DNA, as observed when using M-phase cell extracts (Fig S5). However, although the assembly of Cbf1 and CBF3 did not show an appreciable difference from the M phase (Fig 6A), Cse4, Mif2, and the Ame1-Okp1 complex assembled on CEN DNA at significantly lower levels in G1-phase cell extracts (Fig 6B). Consequently, other kinetochore subunits were reconstituted below the lower limit of quantification (Fig 6C). These findings suggest that robust

assembly of Cse4, besides being the limiting step, is controlled by cell cycle states.

## Discussion

In this study, we describe the development and application of CASA, combining LC-PRM-MS and quantitative concatemers derived from a yeast expression system to measure protein complex stoichiometry, determining the reconstitution efficiencies of kinetochore subunits in concentrated yeast cell lysates. The following discussion details the advantages, caveats, and potential improvements of CASA as revealed through this proof-of-concept biological application.

### Design strategies of the CKP

QconCAT and its variants have been used previously to produce quantitative concatemer standards (Pratt et al, 2006; Scott et al,

**Table 3.  Summary of peptide recovery pre- and post-matrix matching.**

| Peptide | Peptide recovery (%) | | | |
|---|---|---|---|---|
| | After 15 h | | After 30 h | |
| | No matrix | With matrix | No matrix | With matrix |
| Ame1 | 102.2 | 101.5 | 93.4 | 105.9 |
| Cbf1 | 86.5 | 95.6 | 63.9 | 109.8 |
| Cbf2 | 103.9 | 106.0 | 74.1 | 110.1 |
| Cep3 | 82.3 | 105.4 | 55.4 | 105.3 |
| Chl4 | 96.4 | 103.1 | 64.7 | 113.6 |
| Cnn1 | 0.0 | 89.5 | 0.0 | 46.1 |
| Cse4 | 5.4 | 109.6 | 0.1 | 100.7 |
| Ctf13 | 84.4 | 109.1 | 59.4 | 108.0 |
| Ctf19 | 46.0 | 116.9 | 22.5 | 122.0 |
| Ctf3 | 75.2 | 114.0 | 48.1 | 109.6 |
| Dsn1 | 85.8 | 121.6 | 51.0 | 118.9 |
| Hhf1 | 52.9 | 131.9 | 29.2 | 117.5 |
| Hht1 | 93.0 | 114.4 | 64.6 | 124.8 |
| Hta2 | 72.4 | 101.3 | 44.3 | 94.1 |
| Htb2 | 90.7 | 99.7 | 63.7 | 110.3 |
| Iml3 | 95.1 | 94.1 | 74.5 | 104.2 |
| Mcm21 | 103.1 | 102.3 | 86.7 | 113.1 |
| Mif2-1 | 89.1 | 102.9 | 64.0 | 112.8 |
| Mif2-2 | 5.0 | 111.1 | 0.0 | 89.8 |
| Mtw1 | 9.3 | 118.4 | 2.5 | 102.4 |
| Ndc80 | 68.6 | 101.2 | 41.3 | 109.0 |
| Nkp1 | 88.6 | 98.7 | 60.2 | 106.3 |
| Nkp2 | 99.6 | 102.8 | 74.8 | 106.8 |
| Okp1 | 103.9 | 102.1 | 71.6 | 112.6 |
| Spc105 | 88.0 | 100.5 | 60.7 | 99.2 |

Red text with red highlights indicates peptide recovery <70%.

2016; Takemori et al, 2017). Because concatemers are usually unstructured, the expression of CKP in bacteria suffers from protein degradation and the formation of inclusion bodies (Mirzaei et al, 2008). Therefore, a budding yeast protein expression system was selected and implemented with several optimizations to produce CKP. First, a high-copy expression plasmid with a galactose-inducible promoter was used to improve protein yield. Second, an N-terminal GST tag was incorporated to improve CKP solubility and enable absolute quantification (Fig S6). Third, a C-terminal 6xHis-3xFLAG tag was used for detection and purification. Fourth, yeast strains available for SILAC were used to introduce heavy isotope-containing amino acids and generate stable isotope-labeled internal standards (Ong et al, 2002; Chen et al, 2010).

Selecting peptides with the appropriate length and chemical properties is crucial to minimize challenges during method development and eventual quantification (e.g. poor chromatography, peptide stability, poor ionization, interferences). Excessively hydrophilic peptides suffer from poor retention during reverse-phase chromatography, whereas highly hydrophobic peptides are more prone to solubility issues and adsorption losses. For instance, the Htb2 peptide (HAVSEGTR) lacks hydrophobic residues, resulting in poor binding to the C18 column and suboptimal chromatographic separation (Fig S4). This signal loss, compounded by matrix suppression and interferences, hampered precision when this peptide was analyzed in the reconstituted sample (Figs 5 and 6). These issues can be alleviated by choosing a different peptide for quantification or using more internal standards. Hydrophobic peptides, like the Cnn1 peptide (SFLQDLSQVLAR), suffer from stability issues that are likely due to non-specific losses (precipitation/adsorption losses), even in the presence of the stabilization matrix (Fig 3B and Table 3). Peptide length is another important factor to consider, as longer peptides provide a greater diversity of transitions to distinguish the target peptide from background noise and interfering ions. Despite these advantages, practical issues should be considered when selecting longer peptides for concatemer proteins. For example, longer peptides would increase a concatemer's overall length, necessitating multiple concatemers with fewer tryptic peptides in each concatemer to maintain expression and solubility. Although using multiple concatemers increases the complexity of analysis, it is not a limiting factor for CASA because of the relative ease of producing full-length concatemers in the yeast expression system. In addition, though the stability of peptides in two matrices, 0.1% TFA and 50 ng/$\mu$l of digested yeast cytoplasmic proteins, was evaluated, the stability of concatemer proteins before digestion needs to be evaluated in future studies to understand the stability of these proteins and the best storage conditions for long-term use.

Moreover, our results demonstrate that simultaneous monitoring of multiple peptides for each protein of interest is necessary to ensure quantitative accuracy. In this proof-of-concept study, one peptide was chosen for each kinetochore subunit except for Mif2, for which two peptides were chosen. The observed average difference in the quantitative determinations of the two Mif2 peptides was ~34% (Fig 5). Though modest, this discrepancy exceeds the analytical precision of the method. Thus, a couple of additional factors could have contributed to the discordant quantification: first, flanking amino acids surrounding the tryptic sites of each Mif2 peptide were not included in the CKP, which could affect the trypsin digestion efficiency of CKP versus the native Mif2 protein (Benesova et al, 2021). Second, as described above, one of the Mif2 peptides (VAPLQYWR) demonstrated greater instability than its counterpart. Such loss could be non-specific and variable for different peptides. There are two approaches to circumvent this possible loss during sample preparation: the first approach is to characterize the loss of each peptide using the CKP; and the second and more robust approach is to spike the CKP as proteins into the samples before trypsin digestion. For the quantification of reconstituted Mif2, specifically, an orthogonal approach should be considered, such as using AQUA peptide(s) (Gerber et al, 2003) or including another CKP containing additional Mif2 peptides.

Finally, the discrepancy in Mif2 quantification shows that other quantifications made in this study could benefit from additional confirmation, given a single peptide was used for all other proteins. Past studies have recommended using at least three peptides per protein to accurately determine stoichiometry and copy numbers

**Table 4. Summary of CKP peptide measurement linearity and limits of quantification.**

| Range | Peptide | $R^2$ | Slope | Intercept | # of calibrators | LLoQ (pM) | ULoQ (pM) |
|---|---|---|---|---|---|---|---|
| Low | Hta2 | 0.9998 | $1.03 \times 10^{-3}$ | $8.89 \times 10^{-3}$ | 9 | 78 | 60,000 |
| Low | Ctf3 | 0.9999 | $1.89 \times 10^{-3}$ | $-4.49 \times 10^{-2}$ | 9 | 78 | 60,000 |
| Low | Cbf2 | 0.9964 | $9.59 \times 10^{-4}$ | $1.55 \times 10^{-1}$ | 7 | 625 | 60,000 |
| Low | Nkp1 | 0.9966 | $1.03 \times 10^{-3}$ | $7.86 \times 10^{-2}$ | 7 | 625 | 60,000 |
| Low | Iml3 | 0.9991 | $9.25 \times 10^{-4}$ | $5.82 \times 10^{-3}$ | 10 | 78 | 60,000 |
| Low | Htb2 | 0.9999 | $8.34 \times 10^{-4}$ | $1.12 \times 10^{-1}$ | 7 | 625 | 60,000 |
| Low | Mcm21 | 0.9997 | $1.03 \times 10^{-3}$ | $4.15 \times 10^{-3}$ | 10 | 78 | 60,000 |
| Low | Dsn1 | 0.9999 | $1.12 \times 10^{-3}$ | $-2.81 \times 10^{-2}$ | 9 | 78 | 60,000 |
| Low | Mtw1 | 0.9935 | $8.03 \times 10^{-4}$ | $-1.66 \times 10^{-1}$ | 4 | 313 | 2,500 |
| High | Mtw1 | 1 | $1.39 \times 10^{-3}$ | $-1.55 \times 10^{0}$ | 4 | 2,500 | 20,000 |
| Low | Hhf1 | 0.9996 | $7.49 \times 10^{-4}$ | $-2.86 \times 10^{-1}$ | 5 | 1,250 | 60,000 |
| Low | Cbf1 | 0.9971 | $8.05 \times 10^{-4}$ | $3.77 \times 10^{-3}$ | 9 | 78 | 60,000 |
| Low | Cep3 | 0.9952 | $6.99 \times 10^{-4}$ | $6.53 \times 10^{-3}$ | 10 | 78 | 60,000 |
| Low | Ame1 | 0.9987 | $1.12 \times 10^{-3}$ | $-3.23 \times 10^{-2}$ | 10 | 78 | 60,000 |
| Low | Chl4 | 0.9996 | $9.41 \times 10^{-4}$ | $2.32 \times 10^{-2}$ | 10 | 78 | 60,000 |
| Low | Ctf19 | 0.9998 | $1.48 \times 10^{-3}$ | $-8.24 \times 10^{-2}$ | 8 | 78 | 60,000 |
| Low | Ndc80 | 0.9998 | $1.12 \times 10^{-3}$ | $-4.17 \times 10^{-3}$ | 9 | 78 | 60,000 |
| Low | Cnn1 | 0.9903 | $7.32 \times 10^{-4}$ | $-1.44 \times 10^{0}$ | 4 | 2,500 | 20,000 |
| Low | Hht1 | 0.9999 | $8.06 \times 10^{-4}$ | $1.39 \times 10^{-1}$ | 8 | 313 | 60,000 |
| Low | Ctf13 | 0.9995 | $7.87 \times 10^{-4}$ | $2.12 \times 10^{-2}$ | 8 | 156 | 20,000 |
| Low | Mif2-2 | 0.9992 | $1.04 \times 10^{-3}$ | $-2.91 \times 10^{-2}$ | 8 | 78 | 60,000 |
| Low | Spc105 | 0.9999 | $2.43 \times 10^{-3}$ | $-3.13 \times 10^{-2}$ | 7 | 313 | 60,000 |
| Low | Okp1 | 0.9993 | $7.33 \times 10^{-4}$ | $2.99 \times 10^{-2}$ | 9 | 156 | 60,000 |
| Low | Nkp2 | 0.9998 | $7.94 \times 10^{-4}$ | $5.98 \times 10^{-2}$ | 8 | 313 | 60,000 |
| Low | Mif2-1 | 0.9997 | $8.21 \times 10^{-4}$ | $1.57 \times 10^{-1}$ | 6 | 625 | 60,000 |
| Low | Cse4 | 0.9912 | $4.14 \times 10^{-4}$ | $-7.55 \times 10^{-2}$ | 4 | 313 | 2,500 |
| High | Cse4 | 0.9988 | $8.32 \times 10^{-4}$ | $-1.28 \times 10^{0}$ | 5 | 2,500 | 60,000 |

LLoQ, lower limit of quantification; ULoQ, upper limit of quantification.

for bottom-up analysis (Wohlgemuth et al, 2015), which may be financially burdensome for protein complexes with many subunits. This is especially true if isotope-labeled peptides will be used as internal standards. Nevertheless, CASA provides an economical alternative to synthetic peptide standards and could be readily multiplexed to include 2–3 peptides per protein. In summary, future designs of CKPs and applications of CASA should address these limitations, and quantification using more than one peptide specific to each protein should be considered.

### Matrix and CE optimization contribute to optimal analytical performance

The stabilization matrix significantly reduced the recovery losses observed in the more hydrophobic peptides (Fig 3B and Table 3). As peptide recovery losses were non-linear with respect to time, peptide precipitation or non-specific adsorption loss in-vial was the most likely cause of poor recovery (Fig 3). Specifically, matrix stabilization effectively eliminated peptide losses for all except the Cnn1 peptide. As expected, isotope dilution appropriately compensates for the recovery losses of Cnn1, as quantification of the Cnn1 peptide demonstrated acceptable accuracy and precision when isotope diluted, highlighting the need for the use of the stable isotope-labeled internal standard to ensure robust quantification. Interferences from the stabilization matrix were also evaluated, and little to no interference was observed for PRM scans monitoring the light or the heavy peptides of the CKP. The Htb2, Hht1, Hhf1, and Cbf1 peptides have minimal matrix contribution in the light channel with peak areas less than or comparable (Htb2) to the lowest calibrator. Therefore, incorporating a stabilization matrix aided in the overall robustness of the method and did not interfere with the quantification of the 25 targeted peptides. Including the appropriate matrix in CASA as a standard practice is essential to ensure consistent peptide recovery over time.

CE optimization improved the method's sensitivity for most peptides in the CKP (Fig 2C). However, when applied to the analysis

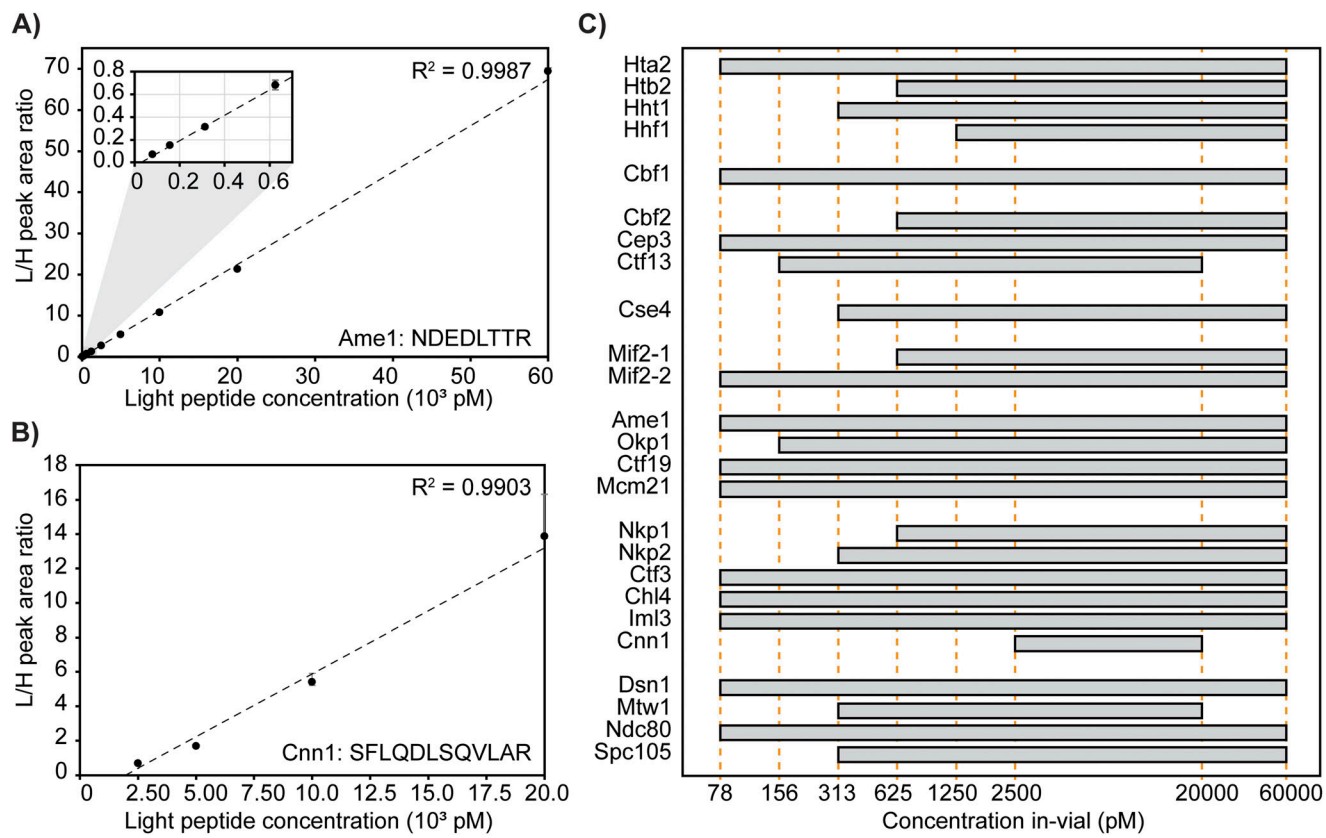

**Figure 4. Linearity and limits of quantification.**
**(A, B)** Representative calibration curve of the Ame1 peptide (NDEDLTTR) and the Cnn1 peptide (SFLQDLSQVLAR). In-vial concentrations (pM) are shown on the x-axis, and the peak area ratio between L-CKP and H-CKP is plotted on the y-axis. A zoom region inset of the plot is appended in the top left corner to show the curve fitting at the lower calibrators. **(C)** AMRs of all CKP peptides: the AMRs of CKP peptides are plotted as gray bars on a $\log_{10}$ scale. Orange dashed lines intersecting the x-axis indicate calibrator concentrations (pM). Dark gray bars indicate the lower AMRs of Cse4 and Mtw1 peptides.
Source data are available for this figure.

of reconstituted kinetochores, interferences were observed for the most abundant product ions for almost all peptides. This stems from specific matrix effects related to reconstitutions performed with CEN, MUT, and ARS DNA; each of these reconstitutions has the potential to have interfering ions unique to the sample. Such occurrences are unavoidable in biological applications and were addressed by changing quantifier ions to those free of interferences. Characterizing optimal CEs for all product ions of each peptide through PRM allowed us to choose optimal CEs in response to changes in quantifier ions. The use of ion ratios adds further robustness by systematically identifying unknown interferences. An alternative method to reduce the effect of interferences on accurate quantification is to use the summed peak areas of the most abundant product ions (generally 3–4 product ions per peptide) (Wohlgemuth et al, 2015). By doing so, occasional interferences in individual product ions are expected to contribute less to the quantification than when single ions are used. To verify that quantification is specific to the analyte of interest, we applied this approach and quantified reconstituted kinetochores using each peptide's top five visible product ions. We observed differences of less than 15% between summed ion quantification and single ion quantification (Fig S7), indicating that both approaches are

equivalent with regard to quantification. Ion ratio monitoring should still be applied to both approaches to ensure the specificity of the analyte being monitored. Moreover, selecting a single quantifier ion would be preferred if the analyte of interest is low in abundance and the sample matrix is highly complex—such as the reconstituted kinetochores—where at the lower ends of the AMR for each peptide, few product ions have adequate signal-to-noise ratios for robust quantification. In this scenario, summing the integrated signal of multiple transition ions may lead to the addition of background noise.

## Advantages of CASA for quantitative proteomics

The proteomics field has produced an enormous amount of qualitative and semi-quantitative data, which is extremely useful for many biological applications. However, conventional proteomics workflows are limited in the settings of stoichiometry analysis, copy-number determination, and absolute protein quantification. The data-dependent acquisition mass spectrometry (DDA-MS) approach, a widely used quantitative proteomics method, performed poorly in the identification of the 25 kinetochore peptides. The analysis of reconstituted kinetochores using DDA-MS only

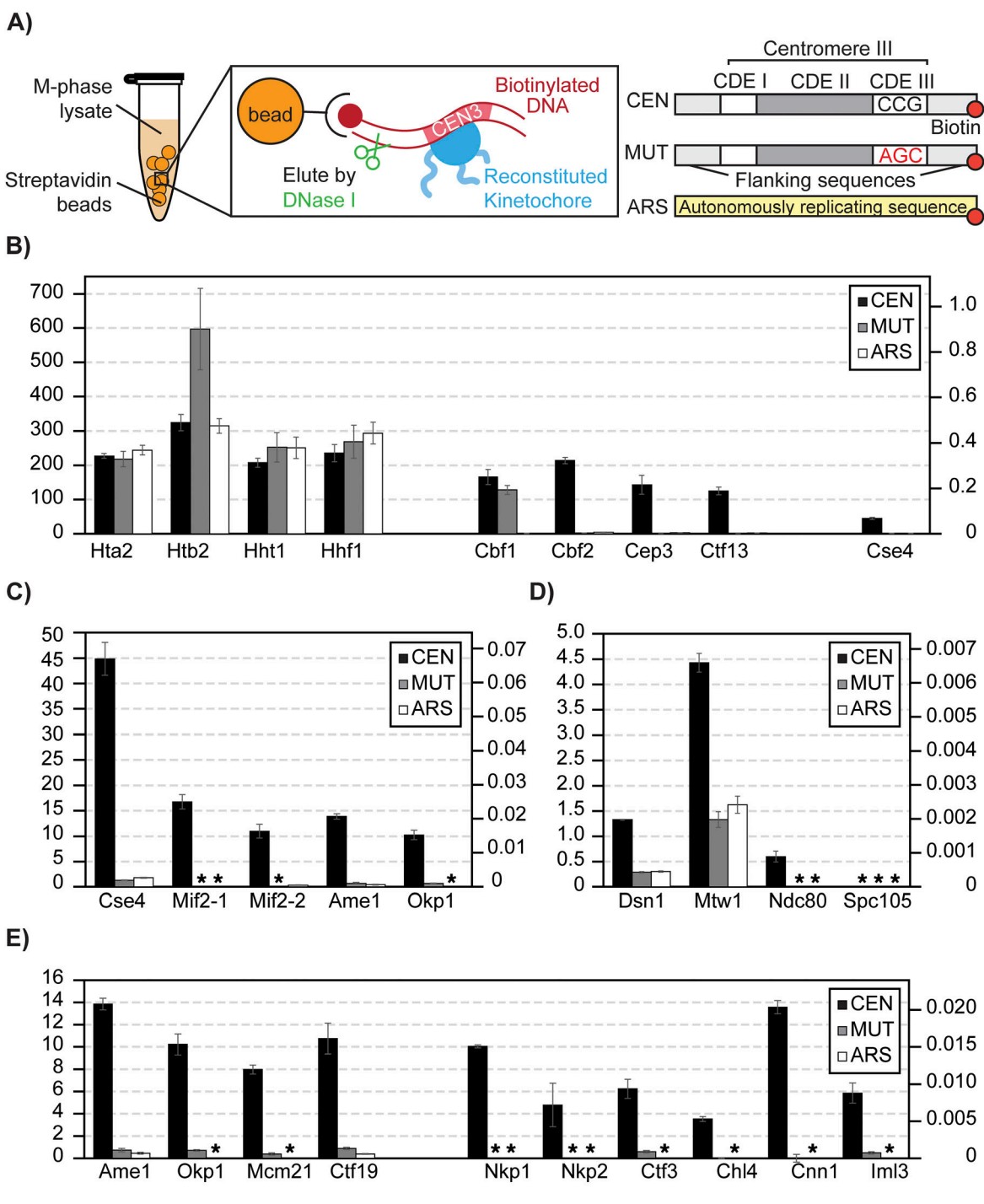

Primary y-axis (left): Normalized concentration (nM)
Secondary y-axis (right): Protein/DNA ratio
*Below limit of quantification

**Figure 5. Stoichiometry of M-phase yeast kinetochores.**
**(A)** Right: schematics of ex vivo kinetochore reconstitution using yeast whole-cell extracts (WCEs); left: DNA maps for the three types of DNA used. **(B)** Quantification of ex vivo reconstituted nucleosomes, Cbf1, and the Cbf3 complex: The primary y-axis (left) shows the normalized concentration of proteins (see the Materials and Methods section). The secondary y-axis (right) shows the amount of each protein as a ratio to the total amount of DNA on beads. The SD of three process replicates (three ex vivo reconstitutions done in parallel) is plotted as error bars around their averages. Quantifications are only made when the measurement falls within the AMRs defined in Fig 4. * indicates signals below the lower limit of quantification. **(C)** Quantifications of ex vivo reconstituted essential yeast inner kinetochore proteins: Cse4, Mif2, Ame1,

identified 8 of the 25 peptides from the biological sample, the spiked internal standard peptides were identified no more than twice per injection, and the Dsn1 and Spc105 peptides were not identified (Table S9). Furthermore, quantification by XPRESS suffered from high variability even in analytical replicates: %CVs for all eight XPRESS values across three analytical replicates exceeded 23%, and the %CVs for Cbf1 and Cnn1 peptides were 38% and 49%, respectively (Table S9). The inherent bias toward more abundant proteins of the DDA workflow and the high susceptibility to interferences of precursor-based quantification are the likely reasons behind the observed variability in XPRESS quantifications. In contrast, the targeted MS using CASA exhibits more robust quantification and should be adopted as the new standard for quantitative proteomics.

Source data are available for this table.

The determination of protein complex stoichiometry remains a significant challenge in proteomics. A few studies have successfully quantified the stoichiometry of protein complexes: the nuclear pore complex (Ori et al, 2013; Kim et al, 2018), the spliceosome complex (Schmidt et al, 2010), the PP2A network (Wepf et al, 2009), the cullin–RING ubiquitin ligase complex (Bennett et al, 2010), and the TNF signalosome (Ciuffa et al, 2022). These are generally either highly abundant and well-behaved complexes, or affinity purification was sufficient to make quantification possible. The reconstituted kinetochore, however, is much more complex than affinity-purified protein complexes, and the kinetochore proteins are very low abundant in the yeast cell lysate. The development of CASA and its application to reconstituted kinetochores have enabled the stoichiometric analysis of low-abundant and highly complex protein complexes. Moreover, a few of the aforementioned studies used AQUA-MS (Bennett et al, 2010; Schmidt et al, 2010; Ciuffa et al, 2022), which is similar in analytical rigor to CASA but much less accessible, with only a few suppliers providing isotope-labeled versions of the peptides at very high costs. Therefore, the development of CASA can facilitate a broader adoption of synthetic protein standards with substantial improvements in ease of use, flexibility, and cost-effectiveness.

## Lessons learned about the kinetochore and future directions

Reconstituted kinetochores using centromeric DNA and concentrated cell extracts are the best-known examples of native centromeric DNA successfully wrapped with Cse4, Mif2, CBF3, and all other essential kinetochore subunits (Lang et al, 2018). However, because of the relatively low abundance of kinetochores in this reconstitution system and the high complexity of the sample derived from cell lysates, the precise stoichiometry of the ex vivo reconstituted kinetochore was not determined (Lang et al, 2018). This study applies CASA to determine the efficiency of ex vivo reconstituted kinetochores and examine how it is influenced by the cell cycle. Based on our observations, canonical histones and DNA proximal factors associated with CEN DNA robustly. The equal loading of canonical histone H3 (Hht1) with the other canonical histones (Hta2, Htb2, Hhf1)

indicates that the Cse4 loading observed was essentially within the error of canonical histone loading. The inefficient loading of Cse4 in cell lysates was highlighted by the observation that Cse4 : Cbf2 or Cse4 : Cep3 ratios were less than 1 : 1, suggesting that Cse4 assembly may be a limiting step in ex vivo kinetochore assembly (Figs 5 and 6). To understand why Cse4 is limiting, follow-up studies evaluating the role of Scm3, the chaperone for Cse4, should be performed (Camahort et al, 2007; Mizuguchi et al, 2007; Stoler et al, 2007). Moreover, the cell cycle dependence of Cse4-loading activity was evident, as approximately 14-fold less Cse4 was recovered on centromeric DNA using G1-phase cell extracts relative to M-phase cell extracts, suggesting the involvement of unknown M-phase signals in promoting Cse4 loading, in line with this step being limiting in this system (Fig 6).

Overall, the inner kinetochore subunits assembled on centromeric DNA more efficiently than the outer kinetochore subunits (Fig 5). This progressive reduction in reconstitution efficiency observed from nucleosome to the outer kinetochore shows that the ex vivo reconstituted kinetochore could be missing factors important for stabilizing the complete kinetochore at each stage of its assembly. Future studies using the kinetochore reconstitution system can be complemented with purified or overexpressed proteins to evaluate missing/limiting factors (Popchock et al, 2023). Such complementation experiments and their evaluation by CASA are essential for understanding the principles of kinetochore assembly. Moreover, it is key to consider that some subunits, particularly those not tightly bound to the centromeric DNA, may be lost during the purification steps. This loss could have led to the observed variation in some of the reported peptide measurements. For example, the Cep3, Iml3, Ndc80, and Nkp2 peptides had %CVs exceeding 15%, likely resulting from slightly different wash conditions among the three process replicates. This indicates that the assembly of these subunits was not as robust under the current conditions used. Such sample loss–associated variation could be remedied by further optimizations of the reconstitution assay, such as the inclusion of the *dsn1-3D* mutant, which was shown to improve outer kinetochore reconstitution (Lang et al, 2018), and the addition/overexpression of Scm3, which could promote Cse4$^{CENP-A}$ nucleosome assembly.

In summary, this study demonstrates the proof-of-concept application of CASA in studying the efficiency of ex vivo reconstituted kinetochores, revealing its potential as a quantitative platform to study native protein complexes in various biological processes.

# Materials and Methods

## The CKP construct

For kinetochore subunits (listed in Table 1), peptide candidates for targeted MS were identified after the analysis by data-dependent

---

and Okp1. **(D)** Quantifications of ex vivo reconstituted outer kinetochore subunits: Dsn1, Mtw1, Ndc80, and Spc105. **(E)** Quantifications of ex vivo reconstituted COMA complex and Ctf3 complex members.
Source data are available for this figure.

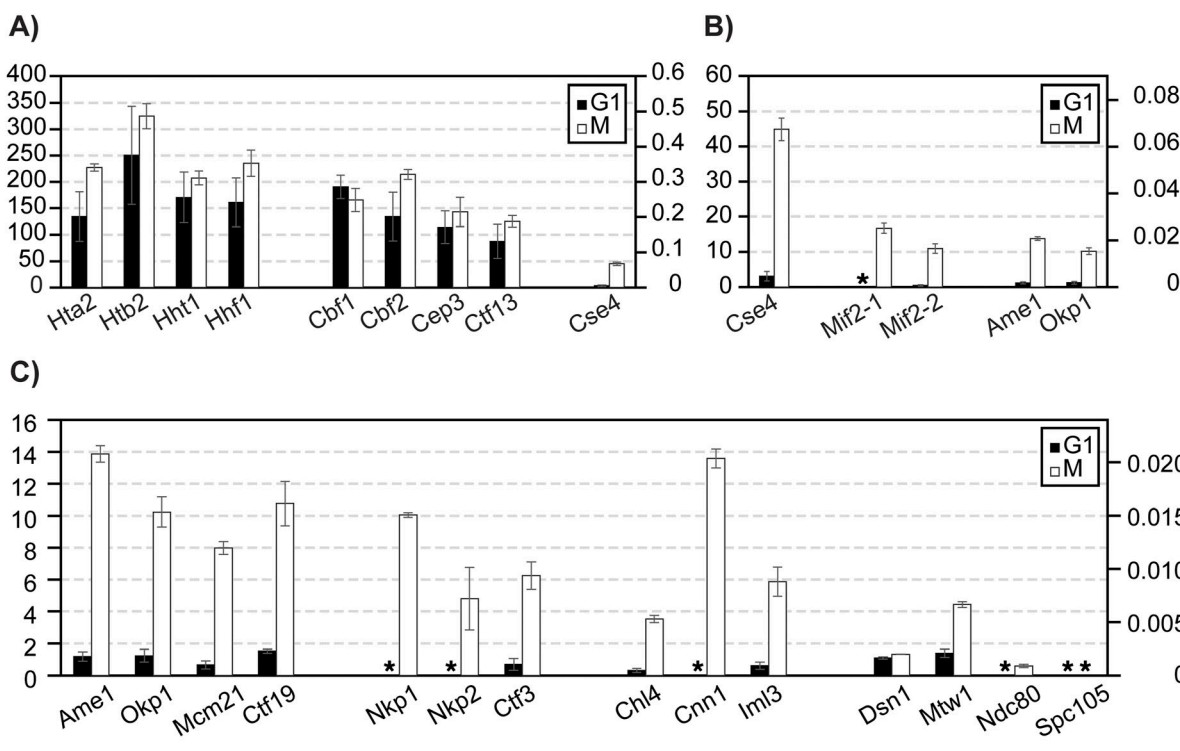

Primary y-axis (left): Normalized concentration (nM)
Secondary y-axis (right): Protein/DNA ratio
*Below limit of quantification

**Figure 6. Effect of cell cycle on ex vivo reconstituted kinetochores.**
**(A, B, C)** Quantifications of ex vivo reconstituted kinetochore subunits when using G1- or M-phase lysates: y-axes are defined as in Fig 5. The plot shows only the results for CEN DNA reconstituted kinetochores. See Fig S10 for MUT DNA data of G1-phase reconstituted kinetochores.
Source data are available for this figure.

acquisition LC-MS/MS of immunoprecipitations of Ame1, Mif2, Cse4, and Ndc80 using Ame1-TAF (HZY2464), Mif2-TAF (HZY2347), 3xFLAG-Cse4 (HZY2777), and Ndc80-TAF (HZY2461) strains (see Table S10, and purification results are shown in Fig S8). Trans-Proteomic Pipeline (TPP, Seattle Proteome Center) (Keller et al, 2005) v6.3.2 Arcus was used to analyze MS data described previously (Suhandynata et al, 2019, 2021). Briefly, MS data were searched using COMET against the Saccharomyces Genome Database (SGD, Stanford University), and peptides were quantified using XPRESS in label-free mode. For database searching, a static modification of 57.0215 D was added for carboxy-amido-methylation of cysteine residues, and a differential modification of 15.9949 D was added for oxidized methionine residues. The final list of targeted peptides was selected from the kinetochore peptides identified by this search (Table S11) based on the integrated signal intensity of the precursor ion, using the following criteria: (1) between 7 and 15 amino acid residues, (2) no methionine or cysteine residues, (3) fully tryptic peptide, (4) ending with arginine, and (5) being unique in the yeast proteome. Quality control was performed for peptide identification through manual inspection of chromatography and MS/MS spectral assignments. A gene block was then designed as follows: an N-terminal flanking sequence of 5'-AATCTATATTTTCAAGGTGGATCCACTAGTTCTAGA-3' (homology to the TEV cleavage site sequence in the plasmid HZE3236),

followed by the sequences of the tryptic peptides lined up head to tail, and finally by a C-terminal flanking sequence of 5'-GGGGGTTCTCAT-CATCATCATCATCATGGGGGCGGA-3' (homology to the 6xHis-3xFLAG sequence in HZE3236). The gene block was codon-optimized and obtained from Integrated DNA Technologies, USA. HZE3236 plasmid linearized by NotI (Cat #: R3189; NEB, England) was repaired by the gene block via homologous recombination in *S. cerevisiae* to make HZE3361. HZE3361 was rescued from yeast genomic DNA by electroporation using electrocompetent *E. coli* cells (in-house–made). A single colony from the rescue was picked and grown in 5 ml Lysogeny Broth (LB) with 100 μg/ml ampicillin to saturation, pelleted, and mini-prepped (GeneJET Plasmid Miniprep Kit, Cat #: K0503; Thermo Fisher Scientific) to obtain the HZE3361 plasmid. HZE3361 was transformed into the yeast strain SCY249 to create yeast strain HZY3059 (Table S10). Galactose induction from this strain led to the expression of the CKP containing a glutathione S-transferase (GST from *Schistosoma japonicum*) tag at the N-terminus and a 6xHis-3xFLAG tag at the C-terminus.

**Isotope labeling and purification of the CKP**

The yeast strain (HZY3059) carrying the pGal-CKP plasmid (HZE3361) was grown at 30°C in 1 liter of synthetic dropout (-His -Arg) media, supplemented with 30 mg/l light arginine ($^{12}C_6$ $^{14}N_4$) or heavy

arginine ($^{13}C_6$ $^{15}N_4$) and 2% raffinose mass/volume (m/v). After cells were grown to saturation, galactose was added to a final concentration of 2% m/v to induce CKP expression for 2 h (Fig S1A). This produced both a light CKP-$^{12}C^{14}N$ and a heavy CKP-$^{13}C^{15}N$, referred to as L-CKP and H-CKP from here onward (Fig S1B). The cells were then harvested and resuspended in 1.5 ml buffer L (25 mM Hepes–KOH, pH 8.0, 175 mM K-glutamate, 2 mM MgCl$_2$, 0.1 mM EDTA, 0.5 mM EGTA, 0.1% NP-40, 15% glycerol m/v, protease inhibitor cocktail, and 1 mM PMSF in deionized H$_2$O). This yeast cell resuspension was flash-frozen in liquid nitrogen drop by drop (popcorn) and stored at –80°C. Before the purification, frozen cell popcorns were lysed using a cryogenic grinder (Cat #: 6875D115; SPEX SamplePrep LLC) at 10 cycles per second for 12 cycles, with a 2-min cycle time and a 2-min cooling time between cycles. The lysed cell powder was thawed on ice and centrifuged at 21,000 RCF for 20 min. 2.5 ml of clarified homogenates was subjected to anti-FLAG-M2 immunoprecipitation with 200 µl of anti-FLAG-M2 agarose resin (Cat #: A2220; Sigma-Aldrich) at room temperature for 1 h. The resin was washed with 5 ml ice-cold buffer L and 5 ml of 0.1% NP-40. The bound proteins were eluted with 1 ml of 0.1% SDS after 1-h incubation at 37°C. The eluate of purified proteins was dried by speed-vac at 55°C and resuspended in 100 µl of 50 mM ammonium bicarbonate in deionized water (dH$_2$O). To remove detergents, proteins were precipitated using 400 µl of 50% acetone/50% ethanol at –20°C overnight. Precipitated proteins were centrifuged at 21,000 RCF for 10 min, and the resulting protein pellet was washed with 5 ml of 40% acetone/40% ethanol/20% dH$_2$O. After washing, the pellet was air-dried (lyophilized) briefly at 37°C to evaporate the remaining organic solvent. The lyophilized CKP was stored at –80°C until use. Trypsin digestion of the protein pellet was performed using 1 µg of modified sequencing grade trypsin (Cat #: V5111; Promega) in 100 µl of 20 mM ammonium bicarbonate at 37°C overnight (12–18 h). Peptides were acidified the next day with 20 µl 10% TFA and diluted to ~100 fmol/µl as a working stock. Working stock aliquots were stored at –80°C until use.

### Absolute quantification of the L-CKP and H-CKP via external calibrations

Absolute quantification of CKP was performed using external calibration with a commercially synthesized peptide (GenScript) of GST at its core region (amino acids 104–108: YGVSR). This GST core peptide is expected in the purified CKPs, which share the N-terminal GST tag and show a lack of a noticeable change in electrophoretic mobility (Fig S1A). Thus, this GST core peptide is iso-stoichiometric with CKP peptides. Possible C-terminal degradation of CKP was also excluded because the 3xFLAG tag at its extreme C-terminus was used to purify the full-length CKP. External calibration was performed by preparing six calibrators of the GST peptide at 625, 1,250, 2,500, 5,000, 10,000, and 20,000 pM in 0.1% TFA. Linear regression with 1/x weighting (calibrators with lower concentrations have more weight) was performed using the integrated peak area of each calibrator (Table S12). All calibrator biases were ±15% from target values, and the R$^2$ value for the GST peptide was 0.9972. Using this external calibration curve, original stocks of L-CKP and H-CKP were determined to be 140 ± 2 nM and 33 ± 5 nM, respectively (Fig S6 and Table S12).

### Validation of absolute quantification using recombinant GST protein

The recombinant GST protein was purified from *E. coli* cells (Rosetta(DE3)pLysS Competent Cells, Cat #: 70956; Sigma-Aldrich) carrying the plasmid HZE2029 (LIC-2GT) using agarose GS-resin (Cat #: 17-5132-02; GE Healthcare) according to the manufacturer's instructions. After purification, GST was precipitated by 50% acetone/50% ethanol and solubilized in 6 M urea/50 mM ammonium bicarbonate in deionized water (dH$_2$O). The concentration of GST was quantified using Beer's law (molar extinction coefficient for GST: $\varepsilon$ = 44,350 cm$^{-1}$ M$^{-1}$) and absorption at 280 nm on a NanoDrop spectrophotometer (Cat #: ND2000; Thermo Fisher Scientific). The GST stock concentration was determined to be 0.91 ± 0.01 µM (Fig S9). This stock was then diluted to 2.5 nM, subjected to trypsin digestion, and quantified using the external calibration curve of the commercially synthesized GST peptide to validate the external calibration curve.

### Reverse-phase liquid chromatography

The liquid chromatography method used two solvents: mobile phase A (MPA, 0.1% formic acid in dH$_2$O) and mobile phase B (MPB, 0.1% formic acid in ACN). Chromatography was obtained using a Vanquish Neo UHPLC with a self-packed fused silica (Cat #: 2000023; Polymicro) C18 column (170 mm length × 100 µm I.D; 2.2 µm particle size, Cat #: 101182-0000; Sepax). Samples were injected using a backward flush Heated Trap-and-Elute workflow (50°C) on a PepMap Neo Trap cartridge (Cat #: 174500; Thermo Fisher Scientific). The analytical column was kept at 50°C using a Sonation PRSO-V2 (Sonation GmbH) column oven. The flow rate was 2.25 µl/min across the entire method, and the initial starting conditions were 98% MPA and 2% MPB. The linear gradient begins at 5.0 min, at which point the composition of MPB increases linearly to 40% across 45.0 min. At 50.0 min, MPB was increased linearly to 95% over 5.0 min, where it was held constant for 5.0 min before returning to 2% MPB over 0.1 min. Equilibration was performed for the final 5.0 min at 2% MPB. The total LC method is 65.1 min, and the injection volume for all analyses was 3 µl.

### PRM-MS

MS analysis was performed in positive ion mode on a Q Exactive Plus mass spectrometer (Thermo Fisher Scientific) using Nanospray Flex Ion Source (Cat #: ES071; Thermo Fisher Scientific). PRM was performed in data-independent acquisition mode with a targeted inclusion list (Table S2) and a single full MS scan accompanying each set of PRM scans. Source parameters are as follows: Spray voltage—2.75 kV, Capillary temperature—290°C, and S-lens RF level—50.0. Full MS parameters are as follows: Resolution—35,000, AGC target—1 × 10$^6$, Maximum IT—50 ms, and Scan range—280–900 m/z. PRM-MS parameters were as follows: Resolution—17,500, AGC target—5 × 10$^5$, and Maximum IT—100 ms. Data were acquired in

Xcalibur (version 4.5.474.0) and analyzed in Skyline (version 23.1) (MacLean et al, 2010; Pino et al, 2020).

## DDA-MS

Comparative studies of the PRM-MS quantification by DDA-MS were performed on a Q Exactive Plus mass spectrometer as above. Source parameters for the DDA method are as follows: Spray voltage—2.75 kV, Capillary temperature—295°C, and S-lens RF level—50.0. Full MS parameters are as follows: Resolution—70,000, AGC target—$3 \times 10^6$, Maximum IT—50 ms, and Scan range—250–1,450 m/z. dd-MS$^2$ parameters were as follows: Resolution—17,500, AGC target—$1 \times 10^5$, Maximum IT—50 ms, TopN—10, Isolation Window—1.0 m/z, and NCE—27. Data were acquired in Xcalibur (version 4.5.474.0) and searched using COMET against the SGD (Stanford University), and peptides were quantified using XPRESS. For database searching, a static modification of 57.0215 D was added for carboxy-amido-methylation of cysteine residues, and differential modifications of 15.9949 D and 10.0083 D were added for oxidized methionine residues and $^{13}C^{15}N$-labeled arginine residues, respectively.

## Evaluation of peptide collision energy

The effect of collision energy on product ion distribution and signal intensities was evaluated by performing replicate injections of ~50 fmol of H-CKP. Across these injections, the collision energy (CE) and NCE were raised by increments of 2% from 10% (lower bound of the instrument) until a Gaussian distribution of chromatographic peak areas was observed for the product ion breakdown curves of each peptide. Integrated peak areas for each peptide's top 10 visible transitions were calculated in Skyline and exported into Excel for subsequent analysis. Optimal collision energies were determined for each peptide based on overall product ion intensities. All breakdown curves are appended as Fig S10. The collision energies used for the PRM-MS method (Table S2) were selected with additional considerations of matrix interferences and quantifier ion signal intensities.

## Generation of stabilization matrix

The matrix used in this study was derived from a tryptic digest of yeast cytosolic proteins obtained through chromatin fractionation, as previously described (Liang & Stillman, 1997). Briefly, 200 OD$_{600}$ ml of yeast cells were treated with lyticase (in-house–purified, 0.5 mg/ml, 800 µl) to generate spheroplasts (Ranish et al, 2004). Spheroplasts were lysed with 0.5% Triton X-100 on ice for 20 min, and the lysate was spun through a 30% sucrose cushion to separate the cytosolic proteins from the chromatin. The soluble fraction (including the sucrose layer) was isolated, reduced with 10 mM dithiothreitol at 37°C for 30 min, alkylated with 30 mM iodoacetamide at room temperature in the dark for 15 min, precipitated by 50% ethanol/50% acetone, and digested into peptides by trypsin. The total protein concentration of the matrix was quantified by the Bradford assay (Bio-Rad). A working stock of the matrix peptides, referred to as matrix

here onward, was prepared at 50 ng/µl in 0.1% trifluoroacetic acid (TFA) and stored at –80°C until used.

## Generation of isotope dilution external calibration curves

Working stocks of calibrators (L-CKP) and the internal standard (H-CKP) were prepared in 50 ng/µl matrix at 120 nM and 2 nM after absolute quantification. The L-CKP was serially diluted to generate a total of 10 working stocks with concentrations ranging from 156.3 pM to 120 nM, one for each calibrator level. Calibrators were prepared in glass inserts (Cat #: 110000101; DWK, GmbH) by mixing 20 µl of the working stocks of L-CKP with 20 µl of the working stock of H-CKP (40 µl final volume). The final concentration of H-CKP (internal standard) in-vial was 1 nM for each of the 10 calibrators, and the final concentration of L-CKP ranged from 78.1 pM to 60 nM. The resulting peak area ratios between L and H-CKP were plotted as a function of L-CKP concentrations in Skyline. Calibration curves were generated by fitting a linear regression with 1/x weighting (calibrators at lower concentrations have more weight) to the plot. Calibrators were excluded if they failed any of the following criteria: (1) signal-to-noise ratio greater than 10, (2) accuracy (%Bias) within ± 20% of target calibrator concentration, and (3) imprecision (%CV) of less than 15% across replicate analysis. Calibration curves generated were only used for downstream quantification if they had an $R^2$ (coefficient of determination) > 0.99 and were fitted to at least four calibrators (had a minimum of four points on the curve). The Mtw1 and Cse4 peptides have four calibration curves that satisfy the set criteria and thus have two overlapping AMRs. The reportable range for each of these two peptides is defined as the range that spans the lowest calibrator of the low-range AMR and the highest calibrator of the high-range AMR. All calibration curves are appended as Fig S11.

## Ex vivo reconstitution of budding yeast kinetochores

### Yeast cell extract preparation

Yeast cells (HZY1028) were grown in 1 liter of YPD (yeast extract, peptone, dextrose; Thermo Fisher Scientific) medium at 30°C to OD$_{600}$ ~ 0.3 and arrested in the M or the G1 phase by adding nocodazole to a final concentration of 150 µg/ml or alpha factor to a final concentration of 15 nM, respectively, for 3 h. After the arrest, cells were centrifuged at 4,000 RCF, and the resulting cell pellet was washed once with PBS (pH 8.0) and resuspended in 1/4 cell pellet volume buffer L. The resuspended cell slurry was flash-frozen in liquid nitrogen and lysed by a cryogenic grinder (Cat #: 6875D115; SPEX SamplePrep LLC) as described above. The protein concentration of the clarified yeast extract was ~80 mg/ml by the Bradford assay (Bio-Rad). The extract was stored at –80°C until reconstitution experiments were performed.

### Preparation of centromeric and control DNA beads

Yeast centromere III (CEN DNA), centromere III with point mutations to its CDE III region (MUT DNA), and yeast autonomously replicating sequence (ARS DNA) were amplified from HZE3240, HZE3241, and HZE3246, respectively, under standard PCR conditions with 0.5 µg Taq polymerase (in-house–purified)/20 mM Tris–HCl (pH 8.4)/50 mM KCl/dNTP (50 µM each)/2 mM MgCl$_2$ in

dH$_2$O and primers 1 and 2 for CEN and MUT DNA, and primers 3 and 4 for ARS DNA: (1) CEN3-For: 5′-GGCGATCAGCGCCAAACA-3′; (2) Bio-CEN3-Rev: 5′-/5Biosg/CGCTCGAATTCGGATCCG-3′; (3) TRP1-For: 5′-GAAGCAGGTGGGACAGGT-3′; and (4) Bio-ARS-Rev: 5′-/5Biosg/CCCCCTGCGATGTATATTTTC-3′. To improve PCR efficiency, additional dTTP and dATP were supplemented (to a final concentration of 200 $\mu$M each) when amplifying the AT-rich CEN and MUT DNA. The resulting PCR product was 411 bps in length. PCR products were precipitated and washed with 70% ethanol/30% 100 mM ammonium acetate in dH$_2$O. The DNA pellet was resuspended in Tris–EDTA buffer (pH 7.5), and the concentrations of the purified DNA were measured by a NanoDrop spectrophotometer (Cat #: ND2000; Thermo Fisher Scientific). Before each reconstitution, 50 $\mu$l of Dynabeads M-280 streptavidin (Cat #: 11206D; Thermo Fisher Scientific) was incubated with 2–3 $\mu$g of the purified DNA at room temperature for 20 min in 1 M NaCl/0.5 nM EDTA/5 mM Tris–HCl (pH 7.5) in dH$_2$O. The amount of DNA bound to the beads was calculated to be ~1.5 $\mu$g per 100 $\mu$l of the original bead solution. After binding with DNA, the beads were washed two times by buffer L and stored at 4°C until use.

### Kinetochore reconstitution for PRM-MS analysis

Frozen yeast extracts were thawed from –80°C on ice before reconstitution experiments. For each kinetochore reconstitution, 50 $\mu$l of DNA beads was incubated with 500 $\mu$l of yeast extract at room temperature for 1 h. After incubation, the beads were washed four times with 500 $\mu$l ice-cold buffer L. Each wash was 500 $\mu$l, and the last two washes were allowed to incubate with the beads at room temperature for 2 min before buffer removal. To elute reconstituted kinetochores, the washed beads were resuspended in 100 $\mu$l buffer L with 0.1 U/$\mu$l TURBO DNase I (Cat #: AM2238, Thermo Fisher Scientific) and incubated at room temperature for 1 h. Eluates were reduced, alkylated, and precipitated by 50% ethanol/50% acetone and prepared for MS analysis as described above. To monitor assay performance, blanks (0.1% TFA), double blanks (50 ng/$\mu$l matrix), blanks with internal standards (H-CKP in matrix), and quality controls (L-CKP and H-CKPs spiked at known concentrations) were included at the beginning and end of each batch of analyzed samples and manually examined to ensure consistency in quantification. Peptide measurements were excluded from analysis if they failed any of the following criteria: (1) calculated concentration within the AMR, (2) a signal-to-noise ratio greater than 10, and (3) accuracy (%Bias) of quantifier: qualifier ion ratio within ±30% from the expected ion ratio.

### Data analysis

All quantification of PRM-MS results was done in Skyline. To extract the correct m/z for the L-CKP and H-CKP peptides, the peptide settings used were as follows.

Digestion: Enzyme—Trypsin [KR|P], Max missed cleavages—0, Background proteomes—open reading frame of yeast genome from SGD, Enforce peptide uniqueness by—Proteins; Prediction: Retention time predictor—none, Use measured retention time when present—TRUE, Time window—5 min; Filter: Min length—5, Max length—25, Exclude N-terminal Aas—0, Exclude potential ragged ends—FALSE, Exclude peptides containing—None, Auto-select all matching peptides—TRUE; Library: None; Modifications: Structural

modifications—Oxidation (M), Max variable mods—3, Max losses—1, Isotope label type—heavy, Isotope modifications—Label: 13C(6) 15N(4) (R), Internal standard type—heavy; Quantification: Regression fit—Linear, Normalization method—Ratio to Heavy, Simple precursor ratios—FALSE, Regression weighting—1/x, MS level—2, Units—pM, Figures of merit—None.

The transition settings used were as described below.

Prediction: Precursor mass—Monoisotopic, Product ion mass—Monoisotopic, Collision energy—None, Declustering potential—None, Optimization library—None, Compensation voltage—None, Use optimization values where present—FALSE; Filter: Precursor charges—1, 2, 3, Ion charges—1, 2, Ion types—b, y, p, From—Ion 1, To—Last ion, Special ions—None, Precursor m/z exclusion window—None, Auto-select matching transitions—TRUE; Library: Ion match tolerance—0.02 m/z, Pick—1 product ions from filtered ion charges and types; Instrument: Min m/z—50 m/z, Max m/z—1,500 m/z, Method match tolerance m/z—0.02 m/z; Full-Scan: Isotope peaks included—Count, Precursor mass analyzer—Orbitrap, Peaks—2, Resolving power—35,000, At—200 m/z, Isotope labeling enrichment—Default, Acquisition method—PRM, Product mass analyzer—Orbitrap, Resolving power—17,500, At—200 m/z, Include all matching scans; Ion Mobility: None.

The peak areas or calculated concentrations of samples were exported from Skyline using the Document Grid. The averages, standard deviations, %CVs, and %Biases were all subsequently calculated in Excel.

Three different amounts of the reconstituted kinetochore samples (0.75%, 7.5%, and 30%) were injected to ensure the peptide measurements were within the reportable range. Normalized concentrations of kinetochore proteins were calculated by dividing the in-vial concentration, as determined by LC-PRM-MS, by the percentage of total sample injected (e.g. 3 nM/7.5% = 40 nM; 3 nM is the measured in-vial concentration, 40 nM is the normalized concentration, and 7.5% is the amount of total sample injected for LC-PRM-MS). To calculate protein: DNA ratios, the calculated concentrations of each peptide were first converted to mols of peptides injected. This value is then normalized against the amount injected (divided by the % injected) to obtain the mols of that peptide within the sample. The mols of DNA used for each reconstitution were calculated using the amount of DNA bound to beads and the molecular weight of the DNA fragment, which was calculated to be ~2 pmol of DNA per reconstitution.

# Data Availability

The mass spectrometry proteomics data have been deposited to the ProteomeXchange Consortium (Deutsch et al, 2023) via the PRIDE (Perez-Riverol et al, 2022) partner repository with the dataset identifier PXD058295.

# Supplementary Information

# Acknowledgements

We thank all members of the Zhou and Suhandynata laboratories at the University of California, San Diego. We thank Dr. Arshad Desai for the critical reading of this article, Dr. Swathi Krishnan at Pfizer, and the Cell Signaling San Diego group for mentoring J Cai. This work was supported by the National Institutes of Health through the following grants: GM151191 (to RT Suhandynata), and GM151191, GM116897, and OD023498 (to H Zhou). J Cai is in addition supported by the Pfizer–Cell Signaling San Diego Fellowship.

## Author Contributions

J Cai: conceptualization, data curation, formal analysis, validation, investigation, visualization, methodology, and writing—original draft, review, and editing.
Y Quan: conceptualization, supervision, methodology, and writing—review and editing.
CY Zhang: investigation.
Z Wang: investigation.
SM Hinshaw: conceptualization, supervision, and writing—review and editing.
H Zhou: conceptualization, resources, data curation, supervision, funding acquisition, investigation, visualization, methodology, project administration, and writing—review and editing.
RT Suhandynata: conceptualization, resources, data curation, supervision, validation, methodology, project administration, and writing—original draft, review, and editing.

## Conflict of Interest Statement

The authors declare that they have no conflict of interest.

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
