## [Reviewer comments · Life Science Alliance]

Life Science Alliance

Concatemer Assisted Stoichiometry Analysis: targeted mass spectrometry for protein quantification

Jiaxi Cai, Yun Quan, Cindy Zhang, Ziyi Wang, Stephen Hinshaw, Huilin Zhou, and Raymond Suhandyata
DOI: <https://doi.org/10.26508/lsa.202403007>

Corresponding author(s): Huilin Zhou, University of California, San Diego and Raymond Suhandyata, University of California, San Diego

Review Timeline:	Submission Date:	2024-08-21
	Editorial Decision:	2024-10-23
	Revision Received:	2024-11-26
	Editorial Decision:	2024-12-02
	Revision Received:	2024-12-06
	Accepted:	2024-12-09

Transaction Report:

October 23, 2024

Re: Life Science Alliance manuscript #LSA-2024-03007-T

Prof. Huilin Zhou
UCSD
Cellular and Molecular Medicine
CMM-East, Room 2070
9500 Gilman Drive
La Jolla, CA 92039

Dear Dr. Zhou,

Thank you for submitting your manuscript entitled "Concatemer Assisted Stoichiometry Analysis (CASA): targeted mass spectrometry for protein quantification" to Life Science Alliance. The manuscript was assessed by an expert reviewer, whose comments are appended to this letter. We invite you to submit a revised manuscript addressing the Reviewer comments.

When submitting the revision, please include a letter addressing the reviewer comments point by point.

Thank you for this interesting contribution to Life Science Alliance. We are looking forward to receiving your revised manuscript.

Sincerely,

-- A letter addressing the reviewer comments point by point.

B. MANUSCRIPT ORGANIZATION AND FORMATTING:

Reviewer #1 (Comments to the Authors (Required)):

The manuscript by Cai et al. presents a novel and innovative approach for determining protein complex stoichiometry through CASA (Concatemer-Assisted Stoichiometry Analysis). This method represents an optimization of the established QConcat technique, where the concatemer protein is expressed in yeast under heavy isotopic labeling conditions, purified via a FLAG tag, and the absolute quantity is accurately measured using a GST tag. The authors have validated their method using heavy and light forms of CKP and subsequently have applied the approach to investigate the stoichiometry of the kinetochore complex under three different DNA forms and throughout two steps of the cell cycle. The application of this method to such a dynamic protein complex like the kinetochore highlights the extensive versatility and potential of CASA in proteomics. I acknowledge the technical rigor and the potential innovation of this approach. However, I outlined some suggestions to enhance the clarity of the manuscript and improve the accessibility of the data generated.

Major point:

- 1) Data acquired must be accessible to the scientific community (e.g. using the PRIDE proteomics exchange database). This includes raw files, Skyline files, Transition results, data analysis scripts and tables. Furthermore, it would be important to report for all experiments, the transition considered for the quantification and how these have been used to calculate the protein amount.
- 2) Improve the clarity of the methodology. A) I suggest the inclusion of a schematic cartoon illustrating all key steps of the CASA quantification (e.g. CKP design, expression of CKP, purification, QC, calculation of the peptide absolute amount. B) The step for peptide quantification should be further clarified. From my understanding the quantification has been performed with only one ion for quantification and one for detection. (S9 and S10 Table). Is it correct? If so, I would recommend considering the use of at least 4/5 transitions for the quantification to have more robust data. C) Could you clarify how the protein/DNA ratio was calculated in your analysis? From the manuscript, I assumed that the DNA component refers to the amount of DNA conjugated to the beads; however, this point is not fully clear.
- 3) In the concatemer protein, only one protein (Mif2) is represented by two peptides, and the authors report distinct behaviours between these peptides. While it may not always be feasible, it is generally recommended to use at least three peptides per protein for accurate determination of stoichiometry and copy numbers (Wohlgenuth et al., *Proteomics*, 2015). Therefore, I suggest calculating the abundance of Mif2 using an orthogonal method, for example using at least three AQUA peptides. This would allow for an evaluation of the observed peptide discrepancies and serve as a benchmark by comparing this approach with another quantitative method (see point 4). Additionally, I recommend including a discussion on the limitations of the current approach, particularly in cases where protein quantification is based on a single peptide.
- 4) When introducing a new method to the scientific community, establishing a benchmark is very important. While I understand that repeating the experiment using the QConcat approach or ordering at least three AQUA peptides for all 25 proteins may not be feasible, I suggest comparing the calculated stoichiometry with the values obtained from DDA acquisition and iBAQ semi-quantitative analysis (Schwanhauser et al. *Nature* 2015).

Minor points.

- 1) NCE optimization. In which matrix was the Normalized Collision Energy (NCE) optimization performed? Could the matrix composition potentially influence the fragmentation efficiency of the peptides? Can the author show all curves of optimization for the 25 peptides?
- 2) Stability. Whether the author has already assessed this point with some data, can the author comment about the stability of the CASA concatemer in -80 stock. For instance, the title of AQUA peptides is granted one year after the purchase. Differently from synthesized peptides, CASA concatemer is longer and trace of proteases during the purification steps could degrade the sample. Could the author comment on this?
- 3) Calibration curve. Like the NCE optimization, I recommend including the calibration curves for all peptides in the supplementary materials for full transparency. In lines 401-403, it is mentioned that "The Mtw1 and Cse4 peptides have two AMRs, which require separate calibration curves (as shown in Figure 4C by the two horizontal bars with different shades of grey)." It seems that two calibration curves are needed due to the LOQ being at 2500 pM. Would it not be more straightforward to simply define the interval where the peptide response is linear up to 2500pM?
- 4) Assessment of the kinetochore stoichiometry. In the case of Mtw1 (Figure 5D), the reported abundance is ~1 nM. Given that only 7.5% of the sample was injected, the measured concentration would correspond to ~75 pM, which is below the LOQ (Table

- 4). If this is correct, Mtw1 would fall below the limit of quantification and, consequently, should not be quantified. Could the authors clarify this point?
- 5) Stoichiometry calculation of CBF3 complex. The authors report a stoichiometry of 12:8:7. However, considering the variability of the measurements, with a coefficient of variation (CV) exceeding 12% for both peptides (Table S11), I would advise caution in proposing a stoichiometry of 8:7 for Cep3 and Ctf13, as this is close to the variability observed in the experiment (Wohlgemuth et al., Proteomics, 2015). It would be beneficial for the authors to comment on how this variability might limit the accuracy of complex stoichiometry calculations.
- 6) The calculation of complex stoichiometry remains a significant achievement in the field of proteomics. Very few studies successfully quantified the stoichiometry of protein complexes, among all the nuclear pore complex (Ori et al MSB 2013, Kim et al Nature 2018), spliceosome complex (Schmidt Anal Chem 2010), the PP2A network (Wepf et al 2009), the Culling-Ring Ubiquitin ligase complex (Bennet et al Cell 2010) and the TNF signalosome (Ciuffa et al PNAS 2021). I suggest including a discussion of this point.
- 7) The authors should discuss the limitations associated with the calculation of stoichiometry from the pulldown of the kinetochore complex. It is key to consider that some subunits, particularly those that are not tightly bound to the complex, may be lost during the purification steps. This loss could significantly impact the accuracy of the stoichiometric measurements.

We would like to thank you for your thoughtful and constructive feedback. We have incorporated your suggestions as outlined below. Please refer to the revised manuscript with all changes marked in blue for your convenience.

“Major point:

1) Data acquired must be accessible to the scientific community (e.g., using the PRIDE proteomics exchange database). This includes raw files, Skyline files, Transition results, data analysis scripts, and tables. Furthermore, it would be important to report the transition considered for the quantification and how these have been used to calculate the protein amount for all experiments.”

Thank you for your kind suggestions. The mass spectrometry proteomics data have been deposited to the ProteomeXchange Consortium via the PRIDE partner repository with the dataset identifier PXD058295. This data is accessible to you before publication with the following access details:

Log in to the PRIDE website using the following details:

Project accession: **PXD058295**

Token: **sAbfm8nl11Vg**

Alternatively, log in to the PRIDE website using the following account details:

Username: **reviewer_pxd058295@ebi.ac.uk**

Password: **cEPnNIAAnWGoq**

Per journal recommendations, we also included the source data files for each figure, which are submitted together with the manuscript using the nomenclature SourceDataFig/TableX.file

The same set of transitions was used for the quantification of all experiments except for the new analysis added in response to comment 2B (for which up to 5 transitions were used to quantify each peptide, and the list of the transitions used is included in **SourceDataFigS11.xlsx**). The information about quantifier and qualifier ions used is listed in **Tables S9 and S10**.

“2) Improve the clarity of the methodology.

A) I suggest the inclusion of a schematic cartoon illustrating all key steps of the CASA quantification (e.g. CKP design, expression of CKP, purification, QC, calculation of the peptide absolute amount).”

We appreciate this remark and have added the following illustration as a graphical abstract to summarize the CASA pipeline: (See next page)

“B) The step for peptide quantification should be further clarified. From my understanding the quantification has been performed with only one ion for quantification and one for detection. (S9 and S10 Table). Is it correct? If so, I would recommend considering the use of at least 4/5 transitions for the quantification to have more robust data.”

This is correct, one transition ion was selected for quantification and another to distinguish interference through ion ratio monitoring. However, we appreciate the comment and have investigated the use of multiple quantitative transition ions.

We have included a figure that illustrates the utilization of the top 5 transition ions for quantification. We compared the quantification of single ion quantification vs multiple ion quantification in **Fig S12**. To summarize, these analyses led to quantifications with % bias < ±15% from the original analysis (only one measurement demonstrated a positive bias of 14%, while all other measurements have % biases < ±11%) (**Fig S12**). As the individual biases are minimal and the mean bias of all measurements was -0.42%, we have kept the original quantification results in the main figures. **Fig S12** and the discussion regarding the different ways to analyze PRM data, such as using multiple transition ions for quantification, have been added to **lines 629-642**.

... An alternative method to reduce the effect of interferences on accurate quantification is to use the summed peak areas of the most abundant product ions (generally 3-4 product ions per peptide). By doing so, occasional interferences in individual product ions are expected to contribute less to the quantification than when single ions are used. To verify that quantification is specific to the analyte of interest, we applied this approach and quantified reconstituted

kinetochores using each peptide's top 5 visible product ions. We observed differences of less than 15% between summed ion quantification and single ion quantification (Fig S11), indicating that both approaches are equivalent with regard to quantification. Ion ratio monitoring should still be applied to both approaches to ensure the specificity of the analyte being monitored. Moreover, selecting a single quantifier ion would be preferred if the analyte of interest is low in abundance and the sample matrix is highly complex – such as the reconstituted kinetochores – where at the lower ends of the AMR for each peptide, few product ions have adequate signal-to-noise ratios for robust quantification. In this scenario, summing the integrated signal of multiple transition ions may lead to the addition of background noise.

“C) Could you clarify how the protein/DNA ratio was calculated in your analysis? From the manuscript, I assumed that the DNA component refers to the amount of DNA conjugated to the beads; however, this point is not fully clear.”

The absolute amount of DNA conjugated to beads was calculated using Nano-drop measurements of the PCR product before and after binding to beads. The calculation is as follows:

$[\text{Before binding (ng/}\mu\text{L)} - \text{after binding (ng/}\mu\text{L)}] \times [\text{volume of DNA solution used for binding (}\mu\text{L)}]$

The amount of each protein (ng) was calculated using the established calibration curves for each peptide and then divided by the % sample injected. For example, for Cse4, we injected 0.75% of the total reconstituted kinetochore eluate to obtain an in-vial concentration of the Cse4 peptide within the reportable range. The calculated in-vial concentration is divided by the volume injected and 0.75% to get the molar amount of Cse4 from one reconstitution reaction. The same analysis is carried out for all peptides in Excel, and the Excel files for all analyses after data export from Skyline are now available both on PRIDE and as source data files with this submission.

The protein/DNA ratio was calculated using the amount (in mols) of DNA and protein.

These details have been added to the method section of the manuscript (**Lines 362-367**).

... To calculate protein : DNA ratios, the calculated concentrations of each peptide were first converted to mols of peptides injected. This value is then normalized against the amount injected (divided by the % injected) to obtain the mols of that peptide within the sample. The mols of DNA used for each reconstitution were calculated using the amount of DNA bound to beads and the molecular weight of the DNA fragment, which was calculated to be ~2 pmol of DNA per reconstitution.

“3) In the concatemer protein, only one protein (Mif2) is represented by two peptides, and the authors report distinct behaviours between these peptides. While it may not always be feasible, it is generally recommended to use at least three peptides per protein for accurate determination of stoichiometry and copy numbers (Wohlgenuth et al., Proteomics, 2015). Therefore, I suggest calculating the abundance of Mif2 using an orthogonal method, for example using at least three AQUA peptides. This would allow for an evaluation of the observed peptide discrepancies and serve as a benchmark by comparing this approach with another quantitative method (see point 4). Additionally, I recommend including a

discussion on the limitations of the current approach, particularly in cases where protein quantification is based on a single peptide.”

Thank you for bringing this point forward; we agree that this is a limitation to the current approach and have discussed the need for multiple peptides to ensure robust quantification of each protein (**lines 591-593** in the revised manuscript). To further clarify this point, we have expanded this section by including more details. Please refer to the newly added **lines 591-601**.

... For the quantification of reconstituted Mif2, specifically, an orthogonal approach should be considered, such as using AQUA peptide(s) or including another CKP containing additional Mif2 peptides.

Finally, the discrepancy in Mif2 quantification shows that other quantifications made in this study could benefit from additional confirmation, given a single peptide was used for all other proteins. Past studies have recommended using at least three peptides per protein to accurately determine stoichiometry and copy numbers for bottom-up analysis, which may be financially burdensome for protein complexes with many subunits. This is especially true if isotope-labeled peptides will be used as internal standards. Nevertheless, CASA provides an economical alternative to synthetic peptide standards and could be readily multiplexed to include 2-3 peptides per protein.

Nevertheless, the variation between two Mif2 peptides does not affect the main conclusions from this work: Cse4 loading is cell cycle controlled and limiting in the cell-free reconstitution of kinetochores.

“4) When introducing a new method to the scientific community, establishing a benchmark is very important. While I understand that repeating the experiment using the QConcat approach or ordering at least three AQUA peptides for all 25 proteins may not be feasible, I suggest comparing the calculated stoichiometry with the values obtained from DDA acquisition and iBAQ semi-quantitative analysis (Schwanhausser et al. Nature 2015).”

Thank you for the suggestion. We have added a DDA-MS analysis of reconstituted kinetochores (**Table S12**). We observed that kinetochore peptide identifications are very low in quality due to the high sample complexity. In such samples, kinetochore subunits are low abundant, so their detection/quantification was often subjected to interferences at the parent ion level.

The addition of isotopically heavy peptide standards allowed us to use the XPRESS quantification algorithm provided by TPP instead of iBAQ to quantify the DDA-MS results; this analysis has been added and discussed in the revised manuscript (**line 649-658**).

*... The analysis of reconstituted kinetochores using DDA-MS only identified 8 of the 25 peptides from the biological sample, the spiked internal standard peptides were identified no more than twice per injection, and the Dsn1 and Spc105 peptides were not identified (**Table S12**). Furthermore, quantification by XPRESS suffered from high variability even in analytical replicates: %CVs for all 8 XPRESS values across 3 analytical replicates exceeded 23%, and the %CVs for Cbf1 and Cnn1 peptides were 38% and 49%, respectively (**Table S12**). The*

inherent bias towards more abundant proteins of the DDA algorithm and the high susceptibility to interferences of precursor-based quantification are the likely reasons behind the observed variability in XPRESS quantifications. In contrast, the targeted MS utilizing CASA exhibits more robust quantification and should be adopted as the new standard for quantitative proteomics.

“Minor points.

1) NCE optimization. In which matrix was the Normalized Collision Energy (NCE) optimization performed? Could the matrix composition potentially influence the fragmentation efficiency of the peptides? Can the author show all curves of optimization for the 25 peptides?”

CE and NCE optimization were carried out using samples dissolved in 0.1% TFA, limiting the amount of matrix effect during CE and NCE optimization.

We have included all breakdown/optimization curves in **Fig S5**.

“2) Stability. Whether the author has already assessed this point with some data, can the author comment about the stability of the CASA concatemer in -80 stock. For instance, the title of AQUA peptides is granted one year after the purchase. Differently from synthesized peptides, CASA concatemer is longer and trace of proteases during the purification steps could degrade the sample. Could the author comment on this?”

This is a great point. We have added a few sentences to the discussion commenting on the stability of the concatemer and the need for further evaluation (**Lines 574-577**).

... Additionally, though the stability of peptides in 2 matrices, 0.1% TFA and 50 ng/μL of digested yeast cytoplasmic proteins, was evaluated, the stability of concatemer proteins before digestion needs to be evaluated in future studies to understand the stability of these proteins and the best storage conditions for long-term use.

Generally, peptides or proteins are best kept as lyophilized powder in aliquots until use. We monitor assay performance by including quality controls at the beginning and end of each batch of samples analyzed. We have detailed this practice in the methods section.

“3) Calibration curve. Like the NCE optimization, I recommend including the calibration curves for all peptides in the supplementary materials for full transparency. In lines 401-403, it is mentioned that “The Mtw1 and Cse4 peptides have two AMRs, which require separate calibration curves (as shown in Figure 4C by the two horizontal bars with different shades of grey).” It seems that two calibration curves are needed due to the LOQ being at 2500 pM. Would it not be more straightforward to simply define the interval where the peptide response is linear up to 2500pM?”

We appreciate this comment greatly and agree that transparency is of utmost importance. These details were not included in the initial submission because we worried about the data being too complex and confusing for most readers. Your comment made us confident that including all the data is the right way to go! All the calibration curves are now included in **Fig S6**.

Regarding the 2 calibration curves of Mtw1 and Cse4 peptides, we kept the lower AMR because the curve satisfies all the analytical criteria and was later used to quantify reconstituted Mtw1 and Cse4. To improve clarity, however, we have combined the 2 AMRs in the main figure. In the method section, we included details about how the AMRs are constructed for these 2 peptides. We hope that including all calibration curves and depositing all data analysis files to the suggested database is sufficient to clarify this point.

“4) Assessment of the kinetochore stoichiometry. In the case of Mtw1 (Figure 5D), the reported abundance is ~1 nM. Given that only 7.5% of the sample was injected, the measured concentration would correspond to ~75 pM, which is below the LOQ (Table 4). If this is correct, Mtw1 would fall below the limit of quantification and, consequently, should not be quantified. Could the authors clarify this point?”

Thank you for bringing this to our attention. A larger % of the sample (30%) was injected to place Mtw1 back into the AMR. We have clarified this by adding an additional section to the methods (added in response to both this comment and comment 1.C.) explaining how data analysis was conducted. **(Lines 330-369)**

“5) Stoichiometry calculation of CBF3 complex. The authors report a stoichiometry of 12:8:7. However, considering the variability of the measurements, with a coefficient of variation (CV) exceeding 12% for both peptides (Table S11), I would advise caution in proposing a stoichiometry of 8:7 for Cep3 and Ctf13, as this is close to the variability observed in the experiment (Wohlgemuth et al., Proteomics, 2015). It would be beneficial for the authors to comment on how this variability might limit the accuracy of complex stoichiometry calculations.”

We appreciate this comment and have incorporated additional discussion at lines **699-708**.

Moreover, it is key to consider that some subunits, particularly those not tightly bound to the centromeric DNA, may be lost during the purification steps. This loss could have led to the observed variation in some of the reported peptide measurements. For example, the Cep3, Iml3, Ndc80, and Nkp2 peptides had %CVs exceeding 15%, likely resulting from slightly different wash conditions among the 3 process replicates. This indicates that the assembly of these subunits was not as robust under the current conditions used. Such sample-loss-associated variation could be remedied by further optimizations of the reconstitution assay, such as the inclusion of the dsn1-3D mutant, which was shown to improve outer kinetochore reconstitution, and the addition/over-expression of Scm3, which could promote Cse4^{CENP-A} nucleosome assembly.

“6) The calculation of complex stoichiometry remains a significant achievement in the field of proteomics. Very few studies successfully quantified the stoichiometry of protein complexes, among all the nuclear pore complex (Ori et al MSB 2013, Kim et al Nature 2018), spliceosome complex (Schmidt Anal Chem 2010), the PP2A network (Wepf et al 2009), the Culling-Ring Ubiquitin ligase complex

(Bennet et al Cell 2010) and the TNF signalosome (Ciuffa et al PNAS 2021). I suggest including a discussion of this point.”

We appreciate this suggestion and have incorporated your points in lines **660-672**.

The determination of protein complex stoichiometry remains a significant challenge in proteomics. A few studies have successfully quantified the stoichiometry of protein complexes: the nuclear pore complex, the spliceosome complex, the PP2A network, the Culling-Ring Ubiquitin ligase complex, and the TNF signalosome. These are either generally highly abundant and well-behaved complexes, or affinity purification was sufficient to make quantification possible. The reconstituted kinetochore, however, is much more complex than affinity-purified protein complexes, and the kinetochore proteins are very low abundant in the yeast cell lysate. The development of CASA and its application to reconstituted kinetochores have enabled the stoichiometric analysis of low-abundant and highly complex protein complexes. Moreover, a few of the aforementioned studies utilized AQUA-MS, which is similar in analytical rigor to CASA but much less accessible, with only a few suppliers providing isotope-labeled versions of the peptides at very high costs. Therefore, the development of CASA can facilitate a broader adoption of synthetic protein standards with substantial improvements in ease of use, flexibility, and cost-effectiveness.

“7) The authors should discuss the limitations associated with the calculation of stoichiometry from the pulldown of the kinetochore complex. It is key to consider that some subunits, particularly those that are not tightly bound to the complex, may be lost during the purification steps. This loss could significantly impact the accuracy of the stoichiometric measurements.”

These are good points. We have addressed them together as part of our response to Minor comment 5. See **Lines 699-708**.

Thank you again for your careful review, time, and patience. We welcome any further suggestions for improvement.

December 2, 2024

RE: Life Science Alliance Manuscript #LSA-2024-03007-TR

Dr Huilin Zhou
UCSD, Ludwig Institute for Cancer Research
Cellular and Molecular Medicine
9500 Gilman Drive
CMM-east room 3050
La Jolla, CA 92093-0653

Dear Dr. Zhou,

Thank you for submitting your revised manuscript entitled "Concatemer Assisted Stoichiometry Analysis: targeted mass spectrometry for protein quantification". We would be happy to publish your paper in Life Science Alliance pending final revisions necessary to meet our formatting guidelines.

- please be sure that the authorship listing and order is correct
- please upload your graphical abstract as a single file, designated as the filetype "graphical abstract"
- please add your figure legends to the main manuscript text
- please add ORCID ID for secondary corresponding author-they should have received instructions on how to do so
- please consult our manuscript preparation guidelines <https://www.life-science-alliance.org/manuscript-prep> and make sure your manuscript sections are in the correct order
- please use the [10 author names, et al.] format in your references (i.e. limit the author names to the first 10)
- you have a figure callout for Figure 3C, but this is not part of the figure or the figure legend-please correct

LSA now encourages authors to provide a 30-60 second video where the study is briefly explained. We will use these videos on social media to promote the published paper and the presenting author (for examples, see <https://docs.google.com/document/d/1-UWCfbE4pGcDdcgzcmiuJI2XMBJnxKYeqRvLLrLS08s/edit?usp=sharing>). Corresponding or first-authors are welcome to submit the video. Please submit only one video per manuscript. The video can be emailed to contact@life-science-alliance.org

A. FINAL FILES:

B. MANUSCRIPT ORGANIZATION AND FORMATTING:

Sincerely,

Reviewer #1 (Comments to the Authors (Required)):

All of my concerns have been addressed and I recommend that the manuscript be published.

December 9, 2024

RE: Life Science Alliance Manuscript #LSA-2024-03007-TRR

Huilin Zhou
University of California, San Diego
Cellular and Molecular Medicine
9500 gilman drive
CMM-east room 2052
La Jolla, CA 92093-0660

Dear Dr. Zhou,

Thank you for submitting your Methods entitled "Concatemer Assisted Stoichiometry Analysis: targeted mass spectrometry for protein quantification". It is a pleasure to let you know that your manuscript is now accepted for publication in Life Science Alliance. Congratulations on this interesting work.

DISTRIBUTION OF MATERIALS:

Again, congratulations on a very nice paper. I hope you found the review process to be constructive and are pleased with how the manuscript was handled editorially. We look forward to future exciting submissions from your lab.

Sincerely,
